# SCORE REPLACEMENT WITH BOUNDED DEVIATION FOR RARE PROMPT GENERATION

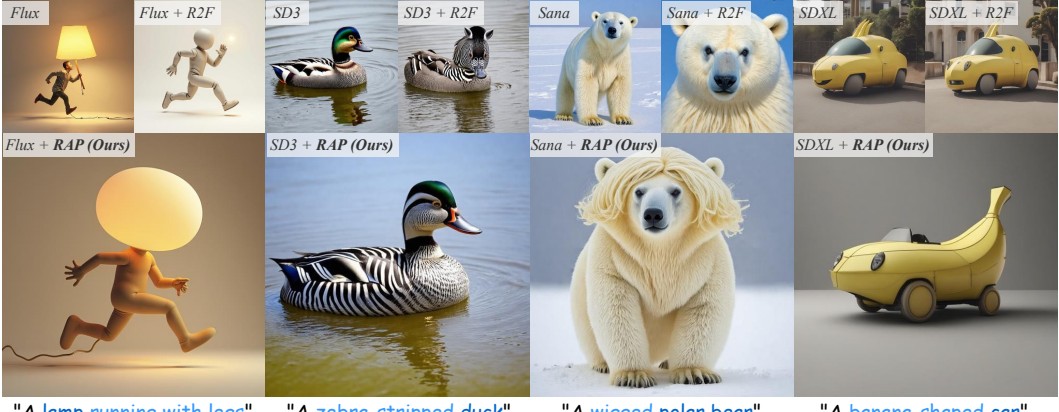

"A lamp running with legs"    "A zebra-stripped duck"    "A wigged polar bear"    "A banana-shaped car"

Figure 1: **Generated samples across different diffusion models for rare concept prompts**. Our method adaptively switches from frequent to rare prompts at the right time. If switching occurs too late, the model aligns only with the frequent part of the prompt (*e.g.*, *lamp running with legs*, *zebra stripped duck*). If switching occurs too early, the rare concept fails to emerge (*e.g.*, *wigged polar bear*, *banana-shaped car*). By adjusting the switching point correctly, our approach produces accurate and consistent realizations of rare concepts across models and settings.

## ABSTRACT

Diffusion models achieve impressive performance in high-fidelity image generation but often struggle with rare concepts that appear infrequently in the training distribution. Prior work attempts to address this issue by prompt switching, where generation begins with a frequent proxy prompt and later transitions to the original rare prompt. However, such designs typically rely on fixed schedules that disregard the model's internal dynamics, making them brittle across prompts and backbones. In this paper, we re-frame rare prompt generation through the lens of **score replacement**: the denoising trajectory of a rare prompt can be initially guided by the score of a semantically related frequent prompt, which acts as a proxy. However, as the process unfolds, the proxy score gradually diverges from the true rare prompt score. To control this drift, we introduce a **bounded deviation** criterion that triggers the switch once the deviation exceeds a threshold. This formulation offers both a principled justification and a practical mechanism for rare prompt generation, enabling adaptive switching that can be widely adopted by different models. Extensive experiments across SDXL, SD3, Flux, and Sana confirm that our method consistently improves rare concept synthesis, outperforming strong baselines in both automated metrics and human evaluations.

## 1 INTRODUCTION

*"The difference between the right word and the almost right word is the difference between lightning and a lightning bug."*

— Mark Twain (1835-1910)

Diffusion models (Rombach et al., 2022; Podell et al., 2023; Chen et al., 2024c;b; Esser et al., 2024; BlackForest, 2024; Liu et al., 2024; Xie et al., 2025) have dominated the generative paradigm, achieving state-of-the-art performance across a wide spectrum of tasks. They excel at high-fidelity image synthesis (Jiang et al., 2024; Zhang et al., 2024; Nie et al., 2025), fine-grained editing (Chen & Wang, 2024; Nguyen et al., 2025), and multi-modal generation (Chen et al., 2024a; Rojas et al., 2025). Their scalability, controllability, and generalization capabilities have driven widespread adoption in both academic and industrial pipelines. However, despite these advances, diffusion models consistently struggle with *rare concepts*, *i.e.*, visual representations of prompts that appear infrequently or are effectively absent in training data (Song & Ermon, 2019). Addressing this limitation is essential for making diffusion models robust to the full diversity of user intent.

A recent effort, Rare-to-Frequent (R2F) (Park et al., 2025), tackles this problem by leveraging large language models (LLMs) to construct semantically related *frequent proxy prompts* (*e.g.*, replacing "a hairy frog" with "a hairy animal"). During generation, the model begins with the proxy prompt to stabilize early steps, and after a predefined number of denoising steps, switches once to the original rare prompt. While effective in some cases, this approach relies on a fixed schedule that is agnostic to the model's internal dynamics. As shown in Figure 1, switching too late causes the model to omit target concepts (*e.g.*, the "lamp" or "duck" in the first two examples), whereas switching too early prevents rare features from emerging (*e.g.*, "wigged" or "banana-shaped" in the last two examples). The lack of principled timing makes R2F brittle across prompts and backbones.

To develop a more effective prompt scheduling strategy, we first revisit the role of prompt switching and interpret it as **score replacement** along the denoising trajectory. This view reveals that while adopting a frequent prompt in the early stage helps enforce semantic attributes, its score function gradually deviates from that of the rare prompt, leading to inferior results if the switch is poorly timed. Building on this perspective, we propose RAP (Rare Concept Generation via Adaptive Prompt Switching), which bounds the score differences to ensure that frequent prompts have sufficient steps to establish semantics, while switching to the rare prompt in time to prevent deviation from the target. In this way, prompt scheduling is no longer a heuristic input manipulation but a structured concept traversal aligned with the model's internal generative process.

In summary, our key contributions are as follows:

- We provide a theoretical foundation for rare concept generation by interpreting prompt switching as score replacement along the denoising trajectory, showing that the score of a rare prompt can be approximated by a semantically related frequent prompt.

- We introduce RAP, an adaptive prompt-switching strategy that bounds score deviations during sampling. This design allows frequent prompts to establish semantic attributes while ensuring timely transitions to rare prompts, leading to faithful rare concept synthesis.

- Extensive experiments demonstrate that RAP consistently improves rare concept generation and outperforms prior methods in both automated evaluations using `GPT-4o` and human preference studies with SDXL, SD3, Flux, and Sana.

## 2 RELATED WORK AND PRELIMINARY

**Text-to-Image Diffusion Models.** Diffusion models (Ho et al., 2020) have emerged as a powerful class of generative models, achieving state-of-the-art performance in text-to-image (T2I) synthesis through increased model capacity and large-scale training data. To address the computational inefficiency of operating in pixel space, Latent Diffusion Models (LDMs) (Rombach et al., 2022) shift the diffusion process into a compressed latent space, enabling high-resolution image generation with reduced cost. Building on this foundation, SDXL (Podell et al., 2023) scales model parameters further and integrates the T5 encoder (Raffel et al., 2020) for improved textual understanding. Transformer-based architectures have also gained popularity for their scalability and performance, as demonstrated in recent works (Peebles & Xie, 2023; Chen et al., 2024c;b). To enhance cross-modal interaction, Multimodal Diffusion Transformers (MM-DiT) (Esser et al., 2024; BlackForest, 2024) concatenate tokens from multiple modalities, enabling joint attention computation across them. Additionally, Sana (Xie et al., 2025) introduces an efficient generation strategy via linear attention, significantly reducing computational overhead while maintaining quality. Despite these advances, generating images from rare or compositional prompts remains a fundamental challenge, often due to poor data

support and unstable latent trajectories. In this work, we build on existing diffusion architectures and propose a framework specifically designed to enhance rare concept generation.

**Preliminary — R2F (Park et al., 2025):** Given a rare input prompt $c_R$, the R2F framework leverages an LLM (*e.g.*, GPT-4o) to construct a sequence of semantically related prompts $\mathbf{C} \triangleq \{c_1, c_2, \ldots, c_n\}$ such that $p_{\text{data}}(c_1) \geq \cdots \geq p_{\text{data}}(c_n)$, with $c_n \equiv c_R$. By construction, the semantic similarity between $c_i$ and $c_R$ increases along the sequence, as shown in Appendix C.1. Alongside these prompts, the LLM predicts a set of switching timesteps $\mathbf{V} \triangleq \{v_1, v_2, \ldots, v_{n-1}\}$ with $v_1 > v_2 > \cdots > v_{n-1}$, where each $v_i$ specifies the step at which the model switches from $c_i$ to $c_{i+1}$. We refer to each transition $c_i \to c_{i+1}$ as **prompt switching**. The sampling process starts from the frequent prompt $c_1$ and, as timesteps decrease from $T$ to 1, gradually progresses toward the rare target prompt $c_R$ following the schedule $\mathbf{V}$. Formally, the active prompt index $i(t)$ at timestep $t$ is $i(t) = \min \{i \in [1, \ldots, n] \mid v_i \geq t\}$. An example of a prompt sequence and switching schedule is provided in Appendix H. To maintain semantic alignment with $c_R$, R2F adopts a prompt alternation strategy. Hence, given $c_t$, the latent $\mathbf{x}_{t-1}$ is updated from $\mathbf{x}_t$ via:

$$\mathbf{x}_{t-1} = \mathbf{x}_t + \text{Denoise}\left(\mathcal{E}_\theta(\mathbf{x}_t, t, c_t), \ t\right), \quad \text{with } c_t \text{ alternate between } \{c_{i(t)}, c_R\}, \tag{1}$$

where $\text{Denoise}(\cdot, \cdot)$ is the denoising function, $\mathcal{E}_\theta$ is the model, $t \in [T, 1]$, and $\mathbf{x}_T \sim \mathcal{N}(\mathbf{0}, \mathbf{I})$.

## 3 RAP Framework

We first conceptualize rare concept synthesis as traversing a *concept trajectory*: a progressive path through the generative space that begins with frequent, well-supported prompts and gradually transitions to rare ones. To move beyond heuristic schedules, we revisit R2F and address its core limitations by: **(i)** providing a theoretical analysis for prompt switching as score replacement during denoising trajectory (Section 3.1); and **(ii)** proposing an adaptive switching rule that bounds the deviation between the concept trajectory and the rare-only trajectory (Section 3.2). Our prompt switching strategy removes the need for heuristic timestep schedules from LLM and instead adapts dynamically to the model, enabling more reliable synthesis of rare concepts.

### 3.1 Prompt Switching as Score Replacement

Rare prompts suffer from poor data support. When a prompt $c_R$ refers to a rare concept, the associated conditional distribution $p_{\text{data}}(\mathbf{x}_t \mid c_R)$ has near-zero density. Such extreme sparsity makes conditioning directly on $c_R$ unreliable, yielding unstable or poorly estimated gradients. In contrast, a semantically related but frequent prompt $c_F$ induces a well-supported distribution $p_{\text{data}}(\mathbf{x}_t \mid c_F)$, allowing more accurate estimation. Prompt switching leverages this by replacing the unreliable guidance from rare prompts $c_R$ with that from frequent prompts $c_F$.

We formalize this mechanism as **score replacement** along the probability flow trajectory. Specifically, let the forward process follow a stochastic differential equation (SDE)[1] defined on the data distribution $p_{\text{data}}(\mathbf{x})$ (Song et al., 2021):

$$d\mathbf{x}_t = \mu(\mathbf{x}_t, t)\, dt + \sigma_t\, d\mathbf{w}_t, \tag{2}$$

where $t \in [0, T]$, $T$ is the maximum timestep, $\mu(\cdot, t)$ is the drift term, and $\sigma_t$ is the diffusion coefficient. This SDE admits an *probability flow* ordinary differential equation (ODE) for reversing, whose solutions at time $t$ follow $p_t(\mathbf{x})$, with $p_0(\mathbf{x}) \equiv p_{\text{data}}(\mathbf{x})$:

$$d\mathbf{x}_t = \left[\mu(\mathbf{x}_t, t) - \frac{1}{2}\sigma_t^2 \nabla_\mathbf{x} \log p(\mathbf{x}_t)\right] dt, \tag{3}$$

where $\nabla_\mathbf{x} \log p_t(\mathbf{x}_t)$ is the *score function* of $p_t(\mathbf{x}_t)$. Under Gaussian probability paths, the drift term and score function are equivalent representations of the dynamics (Holderrieth & Erives, 2025)[2], hence two conditional distributions with similar scores induce similar generative trajectories. This provides a principled view of prompt switching: during early steps, we can approximate the rare prompt score by a frequent proxy score:

$$\nabla_\mathbf{x} \log p(\mathbf{x}_t \mid c_R) \ \approx \ \nabla_\mathbf{x} \log p(\mathbf{x}_t \mid c_F),$$

---

[1]For clarity, we omit conditional inputs.

[2]The details are provided in Appendix A.1

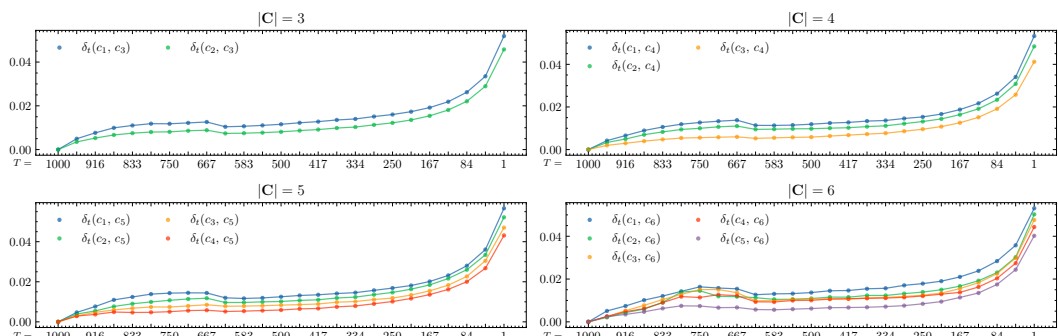

Figure 2: **Score difference between consecutive prompts with Flux** ($T = 1000$ **with 25 inference step).** The $x$-axis denotes the timestep, and the $y$-axis denotes the score difference.

ensuring the denoising path follows a *concept trajectory* that remains close to the rare only path while enjoying the statistical robustness of $c_F$.

To assess the validity of our analysis to prompt switching as score replacement, we measure the score difference between the rare prompt and each frequent prompt. Specifically, during prompt switching, $c_{i+1}$ is applied *after* the latents have been denoised under $c_i$; we mimic this setting when computing the difference. Given a prompt sequence $\mathbf{C} \triangleq \{c_1, c_2, \ldots, c_n\}$ and $c_n \equiv c_R$, the score difference $\delta_t$ between $c_i$ and $c_R$, $\forall i \in [1, 2, \ldots, n-1]$ at timestep $t$ is:

$$\delta_t(c_i, c_R) = \|\nabla_{\mathbf{x}} \log p(\mathbf{x}_t^{c_i} \,|\, c_i) - \nabla_{\mathbf{x}} \log p(\mathbf{x}_t^{c_i} \,|\, c_R)\|_2 \,, \tag{4}$$

where $\mathbf{x}_t^{c_i}$ denotes the latent obtained by denoising with $c_i$ from $t \in [T, 1]$ and $x_T^{c_i} \sim \mathcal{N}(\mathbf{0}, \mathbf{I})$.

Figure 2 reports the average score difference $\delta_t$ for Flux (BlackForest, 2024) under different prompt sequence sizes $|\mathbf{C}|$ using a total of 320 prompts from RareBench (Park et al., 2025).[3] The results indicate that $\delta_t$ is near zero at the earliest steps and increases as noise decays, which reflects the growing influence of prompt semantics on the score. Moreover, prompts that are semantically closer to $c_R$ produce smaller $\delta_t$, supporting the use of smooth prompt sequences that move from frequent to rare. These findings explain why early score replacement is effective, while switching too late risks semantic drift.

The remaining question is *when to switch*. Prior work, such as R2F (Park et al., 2025), adopts fixed schedules provided by LLM between frequent and rare prompts to mitigate drift. Our framework advances beyond these heuristics by treating prompt scheduling as the problem of controlling the accumulated score discrepancy along the denoising trajectory. We introduce an adaptive rule that assigns each prompt a bucket to record the discrepancy and triggers a switch once reaching the bound. This design controls the deviation from the rare-only trajectory and transforms prompt switching from an input-level heuristic into a score-aware traversal.

**Summary.** This section frames prompt switching as score replacement and motivates a score preservation principle. In the early phase, proxy guidance is valid because conditional scores remain close, but as denoising progresses, the discrepancy grows. Thus, switching should be governed by monitored score differences rather than fixed schedules. The adaptive bounded deviation rule in Section 3.2 realizes this principle, yielding a model-agnostic strategy for rare concept synthesis.

## 3.2 ADAPTIVE SWITCHING WITH BOUNDED DEVIATION

As established above, score differences between prompts grow over time and vary with both the backbone and the prompt pair. A fixed schedule assigned by an LLM cannot reliably capture the optimal switching time, which makes such heuristics brittle. Consider the simple case $|\mathbf{C}| = 2$ with $c_1 =$ "a horned animal" and $c_2 = c_R =$ "a horned elephant".

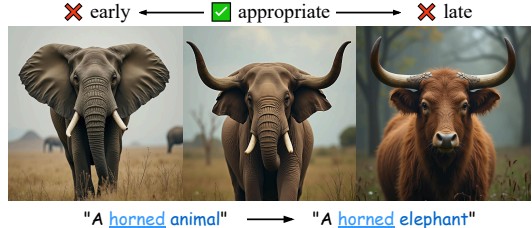

"A horned animal" ⟶ "A horned elephant"

Figure 3: **Prompt switching at different stages**.

---

[3]The corresponding results for other models are provided in Appendix B.

Switching too early prevents the model from retaining the "horned" attribute, while switching too late causes it to miss "elephant." This trade-off, illustrated in Figure 3, motivates an adaptive mechanism that decides when to switch based on the model's internal signals.

From Equation (3), similar score functions yield similar generative trajectories. The design problem is therefore twofold: **(i)** allow the trajectory under a concept sequence $c_\star$ enough steps to imprint semantics, but **(ii)** prevent it from drifting away from the rare prompt guided by $c_R$. Our goal is to establish an upper bound on this concept trajectory deviation and design a switching rule that respects this bound. Formally, let $\mathbf{x}_t^{c_\star}$ denote the trajectory induced under a prompt schedule $c_\star$, where the model applies $c_1$ for $t_1$ steps, then $c_2$ for $t_2$ steps, and so on, ending with $c_R$ for $t_{|\mathbf{C}|}$ steps, with $\sum_i t_i = T$. Let another trajectory $\mathbf{x}_t^{c_\star|R}$ denotes the output obtained by using $c_R$ and $\mathbf{x}_{t+1}^{c_\star}$ to compute the score and serves as the reference trajectory of **what the output would be if the model were conditioned on the original rare prompt $c_R$ for each step** $t$. We then consider the deviation between the final output: $\mathbf{x}_0^{c_\star}$ and $\mathbf{x}_0^{c_\star|R}$, and show that it can be bounded by the accumulated score differences along the trajectory (see Appendix A.2 for detailed formulation and derivation):

$$\|\mathbf{x}_0^{c_\star} - \mathbf{x}_0^{c_\star|R}\| \leq \sum_{i=1}^{|\mathbf{C}|} \sum_{t=t_i^s}^{t_i^e} \kappa_t \, \delta_t(c_i, c_R) \, \Delta t, \tag{5}$$

where $\delta_t(c_i, c_R)$ is the score difference as define in Equation (4), $\kappa_t$ is a schedule-dependent scaling term, and $[t_i^s, t_i^e]$ is the interval assigned to $c_i$.

Inequality (5) shows that the main degree of freedom is the choice of $t_i^s$ and $t_i^e$, *i.e.*, when to switch prompts. We therefore introduce a per-prompt budget that limits the accumulated discrepancy within each segment with a bucket threshold $\delta_B^\star$ and a decay rate $\gamma$:

$$\sum_{t=t_i^s}^{t_i^e} \kappa_t \, \delta_t(c_i, c_R) \, \Delta t \ \leq \ \delta_B^\star \cdot \gamma^{|\mathbf{C}|-i-1}. \tag{6}$$

This construction yields two desirable properties. **(i)** Each frequent prompt is granted enough steps to contribute semantics while keeping deviation controlled. **(ii)** Earlier proxy prompts (*e.g.*, $c_1$) are allowed a smaller budget than later prompts that are closer to $c_R$, which reflects the empirical trend in Section 3.1 that early proxies accumulate larger errors. Summing the budgets across all segments gives a global bound:

$$\|\mathbf{x}_0^{c_\star} - \mathbf{x}_0^{c_\star|R}\| \ \leq \ \sum_{i=1}^{|\mathbf{C}|} \delta_B^\star \cdot \gamma^{|\mathbf{C}|-i-1}. \tag{7}$$

The adaptive switching rule follows directly. For each prompt $c_i$, we maintain an accumulation bucket that sums the scaled score differences over time. Whenever the bucket exceeds its allocated budget $\delta_B^\star \cdot \gamma^{|\mathbf{C}|-i-1}$, the schedule switches to the next prompt $c_{i+1}$. We provide the pseudocode in Algorithm 1 and also demonstrate an example of the score difference at each prompt stage with different colors in Appendix I.

### 3.3 DESIGN DECISIONS

**Bucket Threshold $\delta_B^\star$.** The bucket threshold determines how long a frequent prompt may guide the trajectory before a switch, balancing semantic enrichment and deviation control. Rather than fixing a step budget or tailoring to a specific backbone, we estimate $\delta_B^\star$ from the *early stable regime* of the score discrepancy. Concretely, we detect the first prominent knee (decreasing trend) of $\delta_t$ and define this timestep as $t^\star$, which marks the onset of stable behavior before score deviations begin to rise again. Afterwards, we average deviations up from $T$ to $t^\star$ across sequences:

$$\delta_B^\star = \mathrm{avg}\Big( \big\{ \delta_t(c_i^j, c_R^j) \,\big|\, t \in [T, t^\star], \ i \in \{1, \dots, |\mathbf{C}^j|-1\}, \ \mathbf{C}^j \in \mathbb{C} \big\} \Big), \tag{8}$$

where $\mathbf{C}^j$ is the $j^{th}$ prompt sequence, and $\mathbb{C}$ denotes a set of all prompt sequences. If no clear knee is detected, we directly set $t^\star = 1$, corresponding to using the full trajectory for computing the bucket threshold.[4] Finally, since semantic categories (*e.g.*, shape and texture) often induce distinct $\delta_t$

---

[4]More details are provided in Appendix E.

patterns, we also evaluate category-specific thresholds by applying the same estimator within each class. Ablations in Section 4.4 examine both the global and category-specific settings. In practice, users may not know the prompt category or have a prompt set to compute the group bucket threshold; therefore, we further examine a per-sequence thresholding strategy in Appendix E.3, showing that RAP remains broadly applicable without requiring any predefined concept set.

**Threshold Decay $\gamma$.** Semantically similar prompts tend to accumulate error at similar rates, producing comparable deviation profiles, while dissimilar prompts diverge more quickly. We translate this relation into a decay factor by measuring the semantic similarity between each proxy $c_i$ and the rare prompt $c_R$:

$$\gamma_i = \mathrm{sim}(E(c_i), E(c_R)), \tag{9}$$

where $E$ is a text encoder (T5 (Raffel et al., 2020) or CLIP (Radford et al., 2021)), and sim is cosine similarity. For simplicity, we pre-compute similarities across the entire prompt sequence and use their average as a global decay rate:

$$\gamma = \mathrm{avg}\left(\left\{\mathrm{sim}\left(E(c_i^j), E(c_R^j)\right) \mid i \in \{1, \ldots, |\mathbf{C}^j| - 1\}, \ \mathbf{C}^j \in \mathbb{C}\right\}\right).$$

Importantly, since $\gamma$ depends only on prompt semantics and not on model behavior, it is prompt-specific yet agnostic to the model, and can be applied consistently across different models. We also explore a dynamic strategy that decides the $\gamma$ at runtime and removes the need for a precomputed decay value in Appendix C.2.

# 4 EXPERIMENTS

## 4.1 SETTINGS

**Baselines.** We evaluate our approach against a range of state-of-the-art diffusion models, including SDXL (Podell et al., 2023), SD3 (Peebles & Xie, 2023), Flux.1-dev (BlackForest, 2024), and Sana (Xie et al., 2025). Additionally, we benchmark against R2F (Park et al., 2025), which serves as a strong baseline for rare concept generation. To further assess performance in compositional settings, we include comparisons with attribute-binding and region-controlled methods, including SynGen (Rassin et al., 2023), LMD (Lian et al., 2024), and RPG (Yang et al., 2024). Moreover, on SDXL we additionally benchmark against CADS (Sadat et al., 2024), which enhances sample diversity via prompt perturbation, and MPrompt (Um & Ye, 2025), which targets improved generation of minority and underrepresented concepts.

**Implementation Details.** During inference, we adopt 40 denoising steps for SDXL and 25 steps for the rest and apply classifier-free guidance with the default scale for each model. All experiments are executed on a single NVIDIA H100 GPU using the PyTorch framework. To ensure fair comparison, we reimplemented all baselines in a shared codebase under identical preprocessing, samplers, and hardware, and report the mean over three random seeds. Numbers may differ from those originally reported due to the unified pipeline and seeds.

**Datasets.** We evaluate our framework primarily on **RareBench** (Park et al., 2025), a benchmark designed for rare concept generation. RareBench includes eight categories, covering both single-object and multi-object scenarios with 40 prompts per category. To assess generalization beyond rare concepts, we also report results on **T2I-CompBench** (Huang et al., 2023), which includes six categories with 300 prompts each. Since not all prompts in T2I-CompBench involve rare concepts decided by an LLM, we filter the prompts to have $|\mathbf{C}| \geq 2$, resulting in 268 prompts. Further details on the T2I-CompBench setup are provided in Appendix F.

**Metrics.** Following previous work (Park et al., 2025), we adopt `GPT-4o` as the primary evaluator to assess text-to-image alignment for each generated image. In addition, we measure aesthetic quality using the LAION-Aesthetics Predictor V2.5 model. To complement these automated evaluations, we also conduct user studies on RareBench to validate from the human perception.

Table 1: **Quantitative results on RareBench**. "†" indicates results reproduced by us using multiple random seeds on the same machine. Scores highlighted with a `gray` background are reported from Park et al. (2025).

| Methods | Single Object | | | | | Multiple Objects | | |
| --- | --- | --- | --- | --- | --- | --- | --- | --- |
| | Property | Shape | Texture | Action | Complex | Concat | Relation | Complex |
| SynGen | 61.3 | 59.4 | 54.4 | 33.8 | 50.6 | 30.6 | 33.1 | 29.4 |
| LMD | 23.8 | 35.6 | 27.5 | 23.8 | 35.6 | 33.1 | 34.4 | 33.1 |
| RPG | 33.8 | 54.4 | 66.3 | 31.9 | 37.5 | 21.9 | 15.6 | 29.4 |
| CADS† | 47.3 ± 11.2 | 46.5 ± 11.9 | 57.9 ± 4.1 | 35.2 ± 8.8 | 46.0 ± 1.0 | 26.5 ± 5.3 | 21.7 ± 8.5 | 35.0 ± 4.3 |
| MPrompt† | 47.7 ± 2.9 | 53.1 ± 1.9 | 58.1 ± 9.6 | 51.7 ± 2.2 | 50.8 ± 1.3 | 22.9 ± 5.5 | 22.7 ± 1.9 | 35.4 ± 1.3 |
| SDXL† | 45.0 ± 3.9 | 53.5 ± 7.4 | 59.6 ± 5.8 | 47.5 ± 4.5 | 53.1 ± 3.1 | 29.1 ± 2.6 | 25.4 ± 4.0 | 38.7 ± 3.3 |
| + R2F† | 66.0 ± 6.0 | 62.3 ± 6.7 | 59.0 ± 3.5 | 50.2 ± 2.4 | 58.8 ± 1.1 | 32.5 ± 2.7 | 24.6 ± 4.4 | 39.4 ± 4.1 |
| + RAP | **68.1** ± 6.5 | **64.6** ± 5.1 | **61.9** ± 4.7 | **53.3** ± 3.8 | **59.4** ± 1.7 | **34.0** ± 1.0 | **27.9** ± 2.2 | **40.0** ± 1.9 |
| SD3† | 46.7 ± 3.5 | 69.1 ± 3.6 | 49.4 ± 3.2 | 59.2 ± 2.7 | 62.1 ± 1.3 | 40.4 ± 5.0 | 35.6 ± 0.6 | 57.7 ± 1.4 |
| + R2F† | 71.9 ± 2.3 | 69.6 ± 4.2 | 61.4 ± 3.4 | 67.9 ± 2.9 | 61.3 ± 3.2 | 48.1 ± 1.3 | 41.7 ± 2.0 | 55.6 ± 2.8 |
| + RAP | **72.9** ± 5.2 | 69.6 ± 4.3 | **64.4** ± 4.9 | **68.5** ± 3.6 | **64.2** ± 4.0 | **51.0** ± 5.6 | **42.7** ± 2.5 | **58.1** ± 2.3 |
| Flux.1-dev† | 58.3 ± 6.6 | 59.8 ± 2.0 | 45.2 ± 1.9 | 53.8 ± 3.9 | 56.5 ± 2.8 | 41.7 ± 3.7 | 44.2 ± 1.3 | 56.3 ± 2.7 |
| + R2F† | 57.3 ± 3.8 | 55.0 ± 3.8 | 48.1 ± 6.0 | 44.4 ± 6.0 | 55.0 ± 4.4 | 39.2 ± 3.2 | 41.4 ± 4.2 | 57.3 ± 3.1 |
| + RAP | **67.7** ± 7.8 | **60.2** ± 1.4 | **52.1** ± 5.7 | **54.6** ± 2.5 | **59.6** ± 2.6 | **43.9** ± 2.4 | **46.2** ± 0.7 | **57.9** ± 0.8 |
| Sana† | 61.7 ± 0.3 | 59.0 ± 4.9 | 73.1 ± 1.7 | 69.3 ± 5.0 | 69.8 ± 3.2 | 45.4 ± 7.3 | 36.0 ± 1.3 | 63.5 ± 0.3 |
| + R2F† | 82.1 ± 3.6 | 61.1 ± 6.3 | 73.6 ± 1.6 | 80.8 ± 1.9 | 73.2 ± 1.1 | 53.1 ± 4.4 | 45.2 ± 0.9 | 64.1 ± 1.3 |
| + RAP | **82.9** ± 2.9 | **65.4** ± 3.6 | **74.4** ± 1.9 | **81.0** ± 4.7 | **73.6** ± 1.4 | **54.6** ± 2.0 | **45.6** ± 1.6 | **65.0** ± 1.7 |

Table 2: **Quantitative results on T2I-CompBench**. "†" indicates results reproduced by us using multiple random seeds on the same machine.

| Methods | Color | Shape | Texture | Spatial | Non-Spatial[5] | Complex | **Average** |
| --- | --- | --- | --- | --- | --- | --- | --- |
| SDXL† | 41.2 ± 4.4 | 36.2 ± 0.9 | 51.9 ± 2.0 | 32.2 ± 3.1 | 33.3 ± 14.4 | 26.9 ± 1.9 | 37.0 |
| + R2F† | 38.9 ± 3.5 | 37.1 ± 1.1 | 51.6 ± 2.5 | **36.0** ± 6.8 | 37.5 ± 0.0 | 29.8 ± 4.8 | 38.5 |
| + RAP | **44.2** ± 4.0 | **37.5** ± 4.0 | **53.8** ± 2.5 | 35.4 ± 3.3 | **41.7** ± 7.2 | **30.1** ± 2.0 | **40.4** |
| SD3† | 68.4 ± 2.9 | 61.4 ± 1.7 | 75.6 ± 1.1 | 42.9 ± 2.9 | 37.5 ± 0.0 | 39.1 ± 5.8 | 54.1 |
| + R2F† | 70.3 ± 2.4 | 61.2 ± 3.2 | 72.4 ± 2.0 | 44.0 ± 2.8 | 37.5 ± 0.0 | 40.0 ± 1.1 | 54.2 |
| + RAP | **70.6** ± 1.1 | **61.8** ± 5.0 | **76.4** ± 0.7 | **45.8** ± 2.6 | 37.5 ± 0.0 | **41.7** ± 2.0 | **55.6** |
| Flux.1-dev† | **68.4** ± 1.5 | 55.7 ± 1.3 | 70.8 ± 3.2 | 43.6 ± 2.7 | 37.5 ± 12.5 | 43.3 ± 1.7 | 53.2 |
| + R2F† | 65.0 ± 0.8 | 54.4 ± 3.0 | 66.0 ± 5.1 | 44.7 ± 3.0 | 37.5 ± 12.5 | 35.0 ± 1.1 | 50.4 |
| + RAP | 68.3 ± 2.5 | **58.4** ± 4.0 | **72.2** ± 2.9 | **45.7** ± 3.5 | **41.7** ± 7.2 | **43.9** ± 2.8 | **55.0** |
| Sana† | 55.6 ± 4.2 | 52.5 ± 2.5 | 65.2 ± 3.8 | 41.4 ± 1.7 | 50.0 ± 12.5 | 42.6 ± 2.4 | 51.2 |
| + R2F† | 55.0 ± 1.1 | 52.5 ± 3.8 | 64.9 ± 2.4 | 43.8 ± 1.7 | 50.0 ± 0.0 | 40.1 ± 2.0 | 51.0 |
| + RAP | **55.7** ± 1.1 | **54.3** ± 3.9 | **66.4** ± 3.1 | **48.9** ± 3.0 | 50.0 ± 0.0 | **42.9** ± 1.5 | **53.0** |

## 4.2 MAIN RESULTS

**Quantitative Comparison.** Table 1 presents quantitative results on RareBench across multiple diffusion models. Compared to prior methods targeting attribute binding (SynGen), layout generation (LMD and RPG), prompt perturbation (CADS), and minority improvement (MPrompt), approaches based on the rare-to-frequent mechanism achieve stronger performance on rare concept synthesis. Our method further improves upon R2F, demonstrating the effectiveness of adaptive prompt switching. In particular, for the Flux model, where R2F struggles, our approach achieves gains in all categories. We also evaluate on T2I-CompBench to assess performance on general prompts as shown in Table 2. Here, our method preserves or improves results without degradation for most cases, whereas R2F might underperform on normal prompts for categories such as "Texture" or "Complex". This robustness stems from our adaptive switching rule, which bounds deviation from the original prompt trajectory. Additional results on aesthetic scores are provided in Appendix G.1.

---

[5]This category contains only two samples after filtering, resulting in a large or small standard deviation.

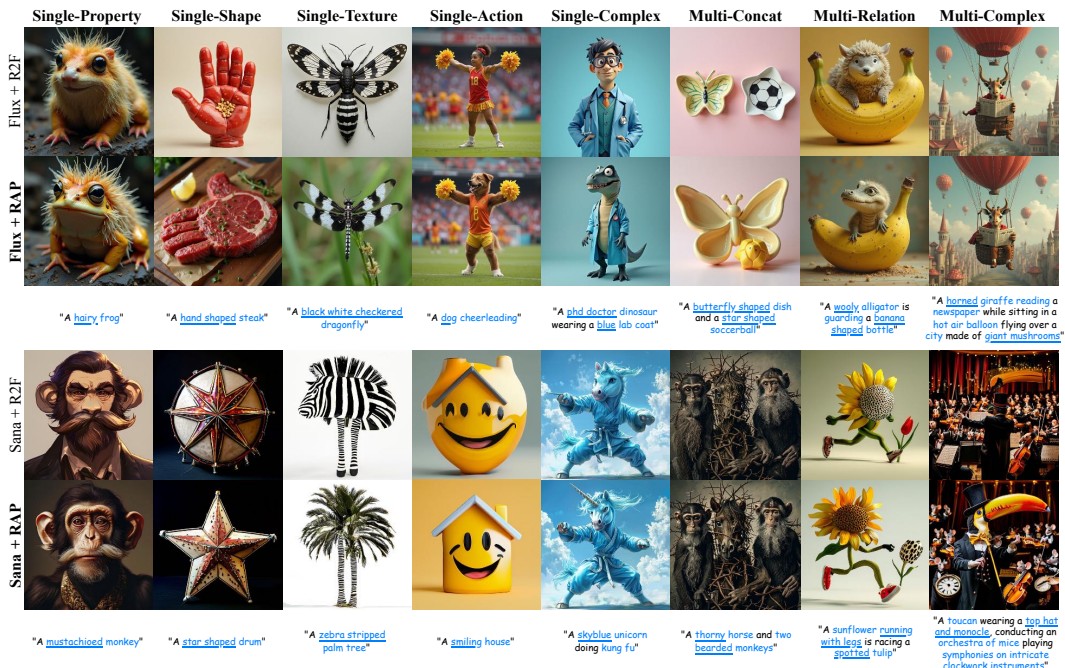

Figure 4: **Qualitative comparison across different models on RareBench for different categories**. The first two rows show results from Flux and the last two from Sana.

**Qualitative Comparison.** Figure 4 shows qualitative results on Flux and Sana across the eight RareBench categories. In several cases, R2F produces inconsistent or semantically incorrect outputs. For instance, Flux+R2F fails to generate the target object ("mushroom") in the "Multi-Complex" category, while Sana+R2F struggles in the "Single-Texture" category, confusing between a "zebra" and a "zebra-stripped" object. In contrast, our method adaptively switches prompts, allocating sufficient steps to frequent prompts while preventing deviation from the rare prompt, thereby producing more faithful and coherent generations. Additional visual results on RareBench and T2I-CompBench are provided in Appendix G.2.

### 4.3 USER STUDY

We conduct a user study with Amazon MTurk to evaluate human preference. The study involves **50 participants**, each shown image-to-image comparisons between our method and baseline approaches. For each case, participants are asked to choose the image based on two criteria: **(i)** text-to-image alignment and **(ii)** overall visual quality. We randomly sample 8 prompts from each RareBench category for each baseline comparison. The win rate shown in Figure 5 is calculated as the percentage of times our method is preferred over the baseline. A win rate above 50% (red horizontal dashed line) indicates that our approach is more frequently favored, high-lighting its perceptual advantage. Full details of the user study setup are provided in Appendix J.

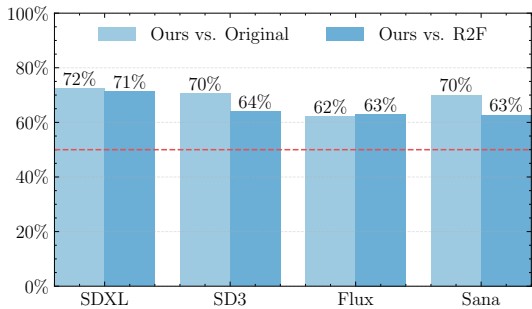

Figure 5: **Results of user study**. $y$-axis shows the win rate of our approach.

### 4.4 ABLATION STUDY

For the ablation study, we use Sana as the target model due to its faster inference, which makes validation more efficient. All experimental settings remain the same with three random seeds.

**Types of Bucket Threshold $\delta_B^\star$.** We further validate that category-specific thresholds can lead to improved performance compared. Quantitative results with Sana are shown in Figure 6, where we only compare categories whose category-specified thresholds differ from the global setting. Using a fixed bucket threshold across all categories yields lower performance compared to thresholds customized by prompt properties. This supports our claim that models respond differently to different types of prompts, and that category-specific thresholds provide a better fit by capturing the distinct prompt patterns.

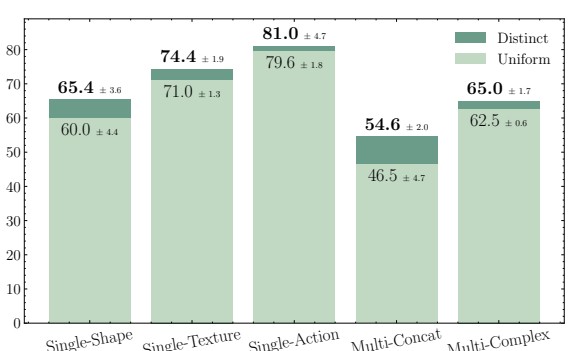

Figure 6: **Comparison of types for $\delta_B^\star$ with Sana.**

**Effect of Decay $\gamma$.** Since $\gamma$ only affects prompt sequences with $|\mathbf{C}| \geq 3$, we evaluate on four categories that fulfill this condition. When $\gamma = 0$, the method degenerates to the original approach, while $\gamma = 1$ removes the decay entirely and treats all prompts equally. As shown in Figure 7 with Sana, introducing decay can improve performance over $\gamma = 1$ under some categories, and setting $\gamma$ to the average semantic similarity (0.9) achieves the best results. Notably, smaller values of $\gamma$ make the behavior resemble the original approach, which performs worst overall. These results confirm that incorporating decay is beneficial, and aligning it with semantic similarity is an effective strategy. We further explore a dynamic $\gamma$ strategy without precomputing a global decay in Appendix C.2.

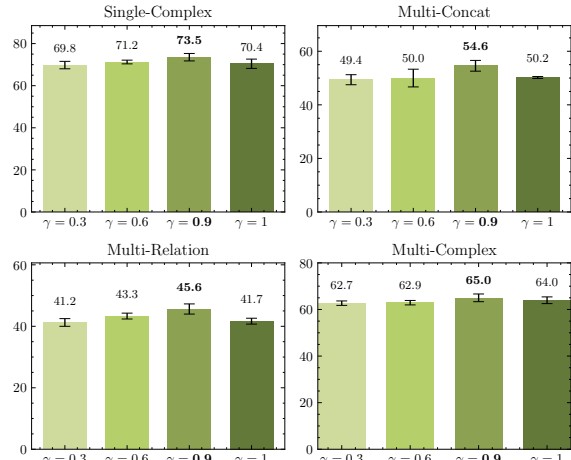

Figure 7: **Comparison of different $\gamma$ with Sana.**

### 4.5 SCORE VISUALIZATION FOR PROMPT SWITCHING

To better understand how different prompt-switching strategies influence the score dynamics, we visualize and compare the average score difference $\delta_t$ between our method and R2F in Figure 8, using Flux as the backbone. Following the setup in Figure 2, we use the full RareBench dataset with prompt-sequence lengths $|\mathbf{C}| \in \{3, 4\}$ and report the average score discrepancy. We remove the prompt alternation and focus only on the stages before the final prompts are applied, as the score difference becomes zero (non-plotted points) once the rare prompt $c_R$ is used. For each setting, we evaluate across the entire RareBench dataset and report the average $\delta_t$ at each timestep.

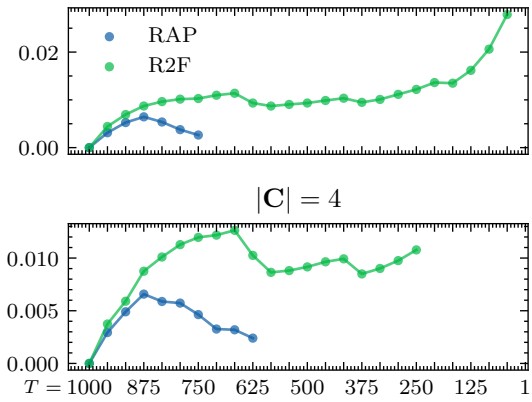

Figure 8: **Comparison of score difference $\delta_t$ between ours and R2F with Flux.** The $x$-axis denotes the inference timestep $T$, and the $y$-axis denotes the score difference $\delta_t$.

From Figure 8, we observe two main trends. First, both approaches show an increasing score difference at the early steps, since frequent prompts carry semantics that differ slightly from the rare prompt, consistent with the pattern observed in Figure 2. Second, as denoising progresses, our method adaptively bounds the score deviation, driving the trajectory closer to the rare-prompt score. In contrast, R2F continues to accu-

mulate error, and $\delta_t$ remains larger, leading to a wider gap from the original rare-prompt trajectory. This comparison highlights how our adaptive strategy preserves score difference more effectively than the fixed schedule in R2F.

### 4.6 RUNTIME AND MEMORY ANALYSIS

We quantify the computational overhead introduced by RAP across four diffusion backbones, SDXL, SD3, Flux, and Sana, in Table 3. All measurements are performed on a single NVIDIA H100 GPU and averaged over three runs. We report inference time (s/img), peak GPU memory, and relative overhead for prompt-sequence lengths $|\mathbf{C}| \in \{2, 5\}$. RAP introduces additional computation because it **(i)** evaluates denoising scores for both the current proxy prompt and the rare target prompt, and **(ii)** computes the score discrepancy $\delta_t$ at each timestep. Importantly, RAP *does not* double the computational cost: once the final rare prompt is reached, the sampling process naturally reverts to the standard single-prompt setting, so dual-score evaluation is required only in the early and mid stages of denoising. This behavior keeps the amortized overhead lower than computing two full trajectories. Memory usage also remains essentially unchanged, as the additional activations and text embeddings contribute negligibly to peak memory. Overall, RAP introduces a modest runtime overhead of approximately 10–30%, depending on the backbone and prompt-sequence length, while leaving memory consumption effectively unchanged.

Table 3: **Runtime and memory analysis of RAP**.

| Model | Time (s) | Peak Mem (GB) | Overhead |
|---|---|---|---|
| SDXL | 2.664 | 8.98 | — |
| + RAP ($|\mathbf{C}|$=2) | 2.978 | 8.98 | +11.8% |
| + RAP ($|\mathbf{C}|$=5) | 3.153 | 8.98 | +18.3% |
| SD3 | 2.336 | 16.91 | — |
| + RAP ($|\mathbf{C}|$=2) | 2.608 | 16.91 | +11.7% |
| + RAP ($|\mathbf{C}|$=5) | 2.802 | 16.93 | +20.0% |
| Flux | 8.123 | 33.85 | — |
| + RAP ($|\mathbf{C}|$=2) | 10.115 | 33.85 | +24.5% |
| + RAP ($|\mathbf{C}|$=5) | 10.623 | 33.86 | +30.8% |
| Sana | 1.040 | 10.18 | — |
| + RAP ($|\mathbf{C}|$=2) | 1.201 | 10.18 | +15.5% |
| + RAP ($|\mathbf{C}|$=5) | 1.441 | 10.19 | +38.6% |

## 5 CONCLUSION AND FUTURE WORK

We introduced RAP, a framework for rare concept generation that interprets prompt switching as score replacement and proposes an adaptive switching rule with a decay mechanism to bound deviations from the rare-prompt trajectory. This design allows frequent prompts to reinforce rare-concept semantics while ensuring outputs remain faithful to the original prompts, enabling stable and coherent synthesis across diverse diffusion backbones. Experiments on SDXL, SD3, Flux, and Sana with both RareBench and T2I-CompBench demonstrate that RAP consistently outperforms prior methods in both automated metrics and human evaluations, yielding more accurate and visually faithful results. Future work includes extending our approach to compositional and unseen concepts, exploring multi-modal frequent-to-rare conditions such as visual and audio prompts, and developing a frequent-to-rare prompt training strategy to improve the model's original ability for rare concept generation.

### REPRODUCIBILITY STATEMENT

We provide our implementation in an anonymous code repository: `https://anonymous.4open.science/r/RAP-CODE`. Details of the experiment settings are included in Section 4.1.

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

# A  FORMULA DERIVATION

## A.1  TRANSFORMATION BETWEEN SCORE FUNCTION AND VECTOR FIELD

Both diffusion and flow-based models employ a Gaussian probability path to interpolate between the initial Gaussian noise distribution and the data distribution. Let $\alpha_t$ and $\beta_t$ be differentiable, monotonic noise schedules satisfying $\alpha_0 = 1$, $\beta_0 = 0$, $\alpha_1 = 0$, and $\beta_1 = 1$. The Gaussian probability path is defined as:

$$p(\mathbf{x}_t \mid \mathbf{x}_0) = \mathcal{N}\big(\mathbf{x}_t;\ \alpha_t \mathbf{x}_0,\ \beta_t^2 \mathbf{I}\big),$$

where $\mathbf{x}_0 \sim p_{\text{data}}$.

Although both diffusion and flow-based approaches share this general formulation, they differ in their training objectives: diffusion models directly align the learned output $\mathcal{E}_\theta$ with the score function $\nabla_{\mathbf{x}} \log p(\mathbf{x}_t)$, whereas flow-based models learn the vector field $\boldsymbol{\mu}(\mathbf{x}_t, t)$. Under the Gaussian probability path, however, these two perspectives are connected, and we can transform between the score and the vector field (Holderrieth & Erives, 2025). Specifically, we have:

$$\boldsymbol{\mu}(\mathbf{x}_t, t) = \left( \beta_t^2 \frac{\dot{\alpha}_t}{\alpha_t} - \dot{\beta}_t \beta_t \right) \nabla_{\mathbf{x}_t} \log p(\mathbf{x}_t) + \frac{\dot{\alpha}_t}{\alpha_t} \mathbf{x}_t \quad \Leftrightarrow \quad \nabla_{\mathbf{x}_t} \log p(\mathbf{x}_t) = \frac{\alpha_t \boldsymbol{\mu}(\mathbf{x}_t, t) - \dot{\alpha}_t \mathbf{x}_t}{\beta_t^2 \dot{\alpha}_t - \alpha_t \dot{\beta}_t \beta_t}, \tag{10}$$

where $\dot{\alpha}_t = \partial_t \alpha_t$ and $\dot{\beta}_t = \partial_t \beta_t$.

Substituting Equation (10) into the denoising formulation Equation (3), we obtain a representation expressed purely in terms of the score function:

$$d\mathbf{x}_t = \left[ \left( \beta_t^2 \frac{\dot{\alpha}_t}{\alpha_t} - \dot{\beta}_t \beta_t - \tfrac{1}{2} \sigma_t^2 \right) \nabla_{\mathbf{x}} \log p(\mathbf{x}_t) + \frac{\dot{\alpha}_t}{\alpha_t} \mathbf{x}_t \right] dt. \tag{11}$$

A symmetric derivation can be obtained using only the vector field $\boldsymbol{\mu}(\mathbf{x}_t, t)$.

In practice, for flow-based models (*e.g.*, Flux, SD3, and Sana), we use Equation (10) to compute the score function. For diffusion models (*e.g.*, SDXL), the conditional score simplifies to:

$$\nabla_{\mathbf{x}} \log p(\mathbf{x}_t) = -\frac{\mathcal{E}_\theta(\mathbf{x}_t, t)}{\beta_t}.$$

## A.2  UPPER BOUND OF THE LATENT DISCREPANCY

We derive here the upper bound in Inequality (5), which connects the score difference between prompts to the latent deviation in the final step:

$$\|\mathbf{x}_0^{c_\star} - \mathbf{x}_0^{c_\star | R}\| \ \leq\ \sum_{i=1}^{|\mathbf{C}|} \sum_{t=t_i^s}^{t_i^e} \kappa_t\, \delta_t(c_i, c_R)\, \Delta t,$$

where $\mathbf{x}_0^{c_\star}$ denotes the output generated under a prompt schedule $c_\star$, and $\mathbf{x}_0^{c_\star | R}$ denotes the output guided by the score computed from $c_R$ and $\mathbf{x}_t^{c_\star}$, with $\mathbf{x}_T^{c_\star | R} = \mathbf{x}_T^{c_\star} \sim \mathcal{N}(\mathbf{0}, \mathbf{I})$:

$$\mathbf{x}_{t-1}^{c_\star} = \mathbf{x}_t^{c_\star} + \text{Denoise}(\mathcal{E}_\theta(\mathbf{x}_t^{c_\star}, t, c_\star),\ t), \tag{12}$$

$$\mathbf{x}_{t-1}^{c_\star | R} = \mathbf{x}_t^{c_\star | R} + \text{Denoise}(\mathcal{E}_\theta(\mathbf{x}_t^{c_\star}, t, c_R),\ t). \tag{13}$$

*Proof.* Recall from Equation (11) that the probability flow ODE can be written as

$$d\mathbf{x}_t = \big[ \kappa_t \nabla_{\mathbf{x}} \log p(\mathbf{x}_t \mid c) + \eta_t \mathbf{x}_t \big] dt, \tag{14}$$

where $\kappa_t$ and $\eta_t$ are deterministic coefficients determined by the noise schedule,

$$\kappa_t = \beta_t^2 \frac{\dot{\alpha}_t}{\alpha_t} - \dot{\beta}_t \beta_t - \tfrac{1}{2} \sigma_t^2, \qquad \eta_t = \frac{\dot{\alpha}_t}{\alpha_t}.$$

Consider two trajectories: **(i)** the concept trajectory induced by prompt schedule $c_\star$ and **(ii)** the concept trajectory induced by $c_R$:

$$d\mathbf{x}_t^{c_\star} = \big[ \kappa_t \nabla_{\mathbf{x}} \log p(\mathbf{x}_t^{c_\star} \mid c_\star(t)) + \eta_t \mathbf{x}_t^{c_\star} \big] dt, \tag{15}$$

$$d\mathbf{x}_t^{c_\star | R} = \big[ \kappa_t \nabla_{\mathbf{x}} \log p(\mathbf{x}_t^{c_\star} \mid c_R) + \eta_t \mathbf{x}_t^{c_\star} \big] dt, \qquad \mathbf{x}_T^{c_\star | R} = \mathbf{x}_T^{c_\star}, \tag{16}$$

where $c_\star(t)$ denotes the active prompt at time $t$ under the schedule $c_\star$.

Subtracting the two ODEs and integrating from $T$ to $0$ gives:

$$\mathbf{x}_0^{c_\star} - \mathbf{x}_0^{c_\star | R} = \int_T^0 \kappa_t \left[ \nabla_{\mathbf{x}} \log p(\mathbf{x}_t^{c_\star} \mid c_\star(t)) - \nabla_{\mathbf{x}} \log p(\mathbf{x}_t^{c_\star} \mid c_R) \right] + \eta_t (\mathbf{x}_t^{c_\star} - \mathbf{x}_t^{c_\star}) dt. \quad (17)$$

Taking norms and applying the triangle inequality yields:

$$\left\| \mathbf{x}_0^{c_\star} - \mathbf{x}_0^{c_\star | R} \right\| \leq \int_T^0 \kappa_t \left\| \nabla_{\mathbf{x}} \log p(\mathbf{x}_t^{c_\star} \mid c_\star(t)) - \nabla_{\mathbf{x}} \log p(\mathbf{x}_t^{c_\star} \mid c_R) \right\| dt + \int_T^0 \left\| \mathbf{x}_t^{c_\star} - \mathbf{x}_t^{c_\star} \right\| dt, \quad (18)$$

where the second term vanishes because the integrand is identically zero. Next, we define the score discrepancy at timestep $t$ as in Equation (4):

$$\delta_t(c_i, c_R) := \left\| \nabla_{\mathbf{x}} \log p(\mathbf{x}_t^{c_\star} \mid c_i) - \nabla_{\mathbf{x}} \log p(\mathbf{x}_t^{c_\star} \mid c_R) \right\|.$$

Partition the trajectory into prompt segments $[t_i^s, t_i^e]$ assigned to $c_i$, $i = 1, \ldots, |\mathbf{C}|$, with $\sum_{i=1}^{|\mathbf{C}|} (t_i^e - t_i^s) = T$. On each segment we have $c_\star(t) = c_i$, so the integrand in Equation (18) becomes $\kappa_t \, \delta_t(c_i, c_R)$. Thus:

$$\left\| \mathbf{x}_0^{c_\star} - \mathbf{x}_0^{c_\star | R} \right\| \leq \sum_{i=1}^{|\mathbf{C}|} \int_{t_i^s}^{t_i^e} \kappa_t \, \delta_t(c_i, c_R) \, dt.$$

Finally, discretizing each integral with step size $\Delta t$ gives:

$$\left\| \mathbf{x}_0^{c_\star} - \mathbf{x}_0^{c_\star | R} \right\| \leq \sum_{i=1}^{|\mathbf{C}|} \sum_{t=t_i^s}^{t_i^e} \kappa_t \, \delta_t(c_i, c_R) \, \Delta t,$$

which corresponds to Inequality 5. $\qquad\square$

## B MORE RESULTS OF SCORE DIFFERENCE

We provide additional results for the score difference $\delta_t(c_i, c_R)$, $\forall i \in [1, \ldots, n-1]$ in Figures 9 to 11 using SDXL, SD3, and Sana. Across all three models, we observe consistent trends: the score difference is near zero at the earliest timesteps, indicating that the noisy latents are largely insensitive to the specific prompt. As the timestep increases and the noise level decreases, $\delta_t$ rises, reflecting the growing influence of prompt-specific semantics on the score function. Moreover, for all $|\mathbf{C}|$, semantically closer prompts to $c_R$ produce more similar scores and have lower score differences. These results align with our main findings in Section 3.1, confirming that the observed behavior of score differences is not specific to a single model architecture.

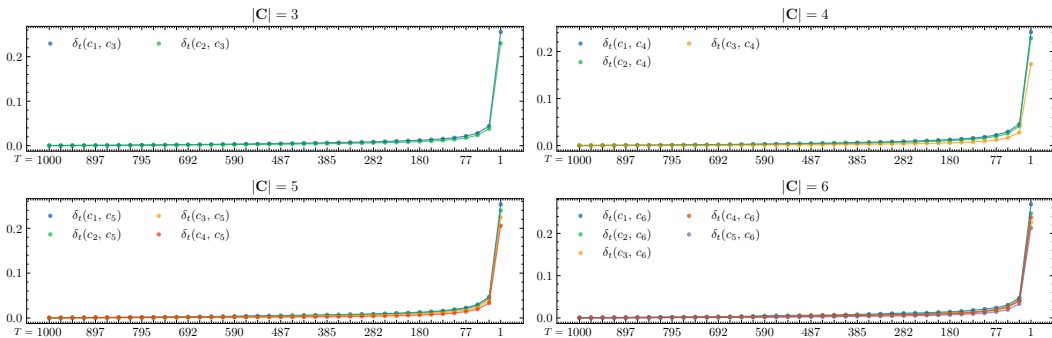

Figure 9: **Score difference between consecutive prompts with SDXL ($T = 1000$ with $40$ inference step).** The $x$-axis denotes the timestep, and the $y$-axis denotes the score difference.

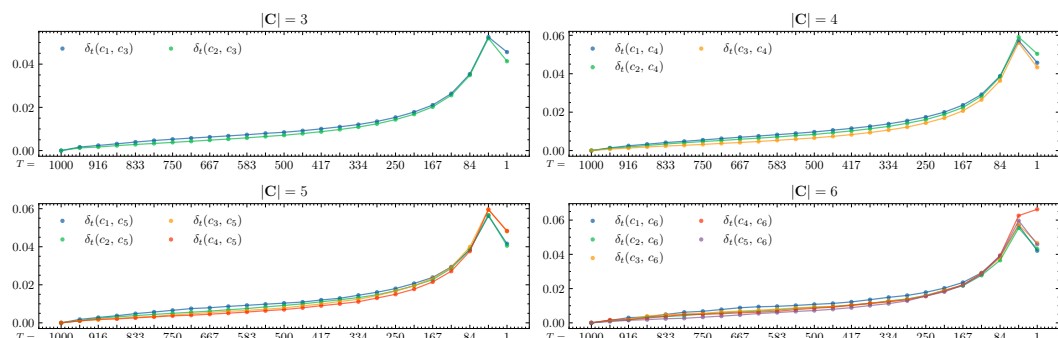

Figure 10: **Score difference between consecutive prompts with SD3 ($T = 1000$ with $25$ inference step).** The $x$-axis denotes the timestep, and the $y$-axis denotes the score difference.

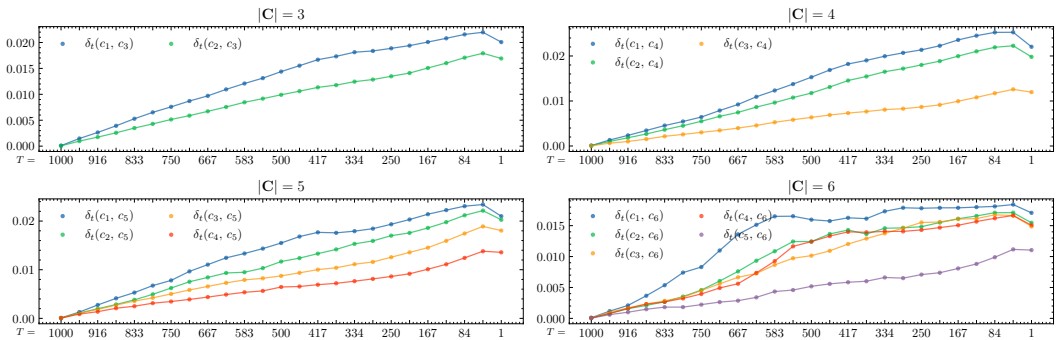

Figure 11: **Score difference between consecutive prompts with Sana ($T = 1000$ with $25$ inference step).** The $x$-axis denotes the timestep, and the $y$-axis denotes the score difference.

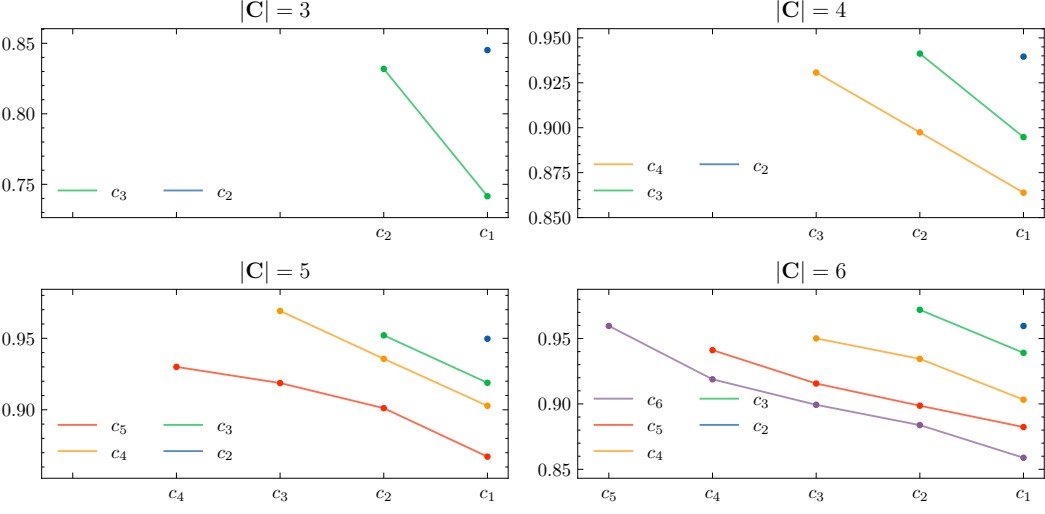

Figure 12: **Text similarity under varying $|\mathbf{C}|$.**

## C  PROMPT ANALYSIS

### C.1  SEMANTIC SIMILARITY

Since the prompt sequences in RareBench are generated by an LLM, we verify that they exhibit a smooth progression and high semantic continuity. Let $\mathbf{C} \triangleq \{c_1, c_2, \cdots, c_n\}$ denote a sequence

where $c_n \equiv c_R$. We compute the pairwise average similarity between prompts $c_i$ and $c_j$ for all $i < j \in [1, \cdots, n]$ under different sequence lengths $|\mathbf{C}|$ (see Figure 12). Each prompt is encoded with the T5 encoder (Raffel et al., 2020) $E(\cdot)$, and similarities are measured by cosine similarity, *i.e.*, $\text{sim}(E(c_i), E(c_j))$. Across all different $|\mathbf{C}|$, the similarity between prompts decays gradually along the sequence, indicating that the LLM transitions smoothly from frequent to rare concepts without abrupt semantic shifts. Additionally, the maximum similarity in each sequence is around 0.95, confirming that consecutive prompts share substantial semantic content. This smooth semantic progression suggests that the LLM-generated sequences are suited for prompt switching.

## C.2 DYNAMIC THRESHOLD DECAY

As discussed in Section 3.3, we can adopt Equation (9) to compute $\gamma$ dynamically, allowing us to rewrite Equation (7) as:

$$\|\mathbf{x}_0^{c_\star} - \mathbf{x}_0^{c_R}\| \leq \sum_{i=1}^{|\mathbf{C}|} \delta^\star \cdot \text{sim}(E(c_i), E(c_R)). \tag{19}$$

This formulation enables the decay to be computed on the fly, without relying on predefined or precomputed values. Results in Figure 13 show that using a dynamic threshold achieves even better performance on both SDXL and Sana in several cases, suggesting that the dynamic threshold offers a more flexible and effective alternative to a fixed decay. We report results on one flow-based model and one diffusion model for faster validation. Since prompt semantics is model-agnostic, the same trends hold across other models of the same type.

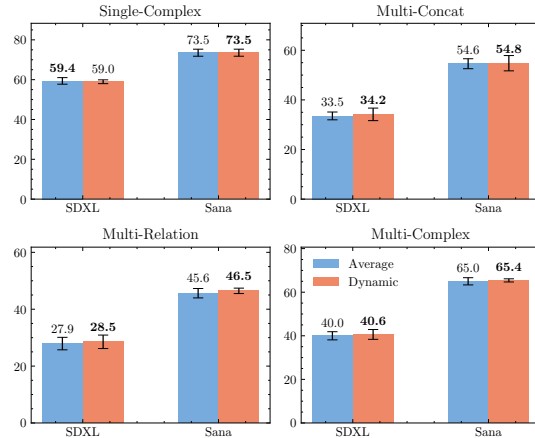

Figure 13: **Comparison of different $\gamma$ strategies.**

## D   RAP ALGORITHM

We present the RAP algorithm in Algorithm 1. Given a prompt sequence $\mathbf{C}$, RAP computes the score difference $\delta_t$ at each step and maintains an accumulation bucket. When the bucket for prompt $c_i$ exceeds its threshold $\tau_i$, the method switches to the next prompt, repeating until the rare prompt is reached. To further preserve score alignment, our approach also incorporates prompt alternation. The decay $\gamma$ can be set either to the average semantic similarity across prompts or computed dynamically using Equation (9).

---

**Algorithm 1:** RAP

---

**Input:** Prompt sequence $\mathbf{C} \triangleq \{c_1, \ldots, c_n\}$ with $c_n \equiv c_R$; total timestep $T$; coefficient $\kappa_t$; step size $\Delta t$; Model $\mathcal{E}_\theta$ ; bucket threshold $\delta_B^\star$; decay factor $\gamma$

**Output:** Final sample $\mathbf{x}_0$

**Assign per-prompt budgets:** $\tau_i \leftarrow \delta_B^\star \cdot \gamma^{\,n-i-1}$, for $i = 1, \ldots, n-1$

Initialize $\mathbf{x}_T \sim \mathcal{N}(\mathbf{0}, \mathbf{I})$, $i \leftarrow 1$, $B \leftarrow 0$;

**for** $t = T, T-1, \ldots, 1$ **do**
    Compute score gap: $\delta_t \leftarrow \|\nabla_\mathbf{x} \log p(\mathbf{x}_t \mid c_i) - \nabla_\mathbf{x} \log p(\mathbf{x}_t \mid c_R)\|$;
    Update bucket: $B \leftarrow B + \kappa_t \cdot \delta_t \cdot \Delta t$;
    **if** $i < n$ *and* $B \geq \tau_i$ **then**
        $i \leftarrow i+1, \quad B \leftarrow 0$;
    **end**
    Select active prompt: $c_t \leftarrow c_i$ ;              // alternate with $c_R$
    $\mathbf{x}_{t-1} \leftarrow \mathbf{x}_t + \text{Denoise}(\mathcal{E}_\theta(\mathbf{x}_t, t, c_t), t)$;
**end**
**return** $\mathbf{x}_0$

---

# E  BUCKET THRESHOLD

## E.1  AVERAGE TO KNEE

As discussed in Section 3.3, we empirically find that computing the bucket threshold up to the first apparent knee (decreasing trend) yields better results. If no knee is observed, we fall back to using the full trajectory. This can be explained by a stable regime: averaging within this regime captures stable behavior, whereas including later steps incorporates rising deviations that overestimate the tolerance. In practice, we observe this phenomenon only in Flux. As shown in Figure 2, the score difference first increases during the early denoising steps and then drops around the $10^{th}$ step. Other models do not exhibit such a clear stable regime and therefore use the full average threshold. A deeper theoretical analysis of these dynamics is left as future work. We also compare the two strategies in Table 4 with Flux, which confirms that averaging up to the knee provides better performance.

Table 4: **Comparison of partial and full averaging strategies on Flux.** Flux-F denotes the full-average threshold, while Flux-P denotes the partial-average threshold computed up to the knee point.

| Methods | Single Object | | | | | Multiple Objects | | |
|---------|----------|-------|---------|--------|---------|--------|----------|---------|
| | Property | Shape | Texture | Action | Complex | Concat | Relation | Complex |
| Flux-F | $67.5 \pm 7.6$ | $55.4 \pm 1.9$ | $50.0 \pm 0.0$ | $48.9 \pm 3.7$ | $58.5 \pm 1.3$ | $39.5 \pm 2.9$ | $42.7 \pm 1.9$ | $55.0 \pm 3.3$ |
| Flux-P | $\mathbf{67.7} \pm 7.8$ | $\mathbf{60.2} \pm 1.4$ | $\mathbf{52.1} \pm 5.7$ | $\mathbf{54.6} \pm 2.5$ | $\mathbf{59.6} \pm 2.6$ | $\mathbf{43.9} \pm 2.4$ | $\mathbf{46.2} \pm 0.7$ | $\mathbf{57.9} \pm 0.8$ |

## E.2  BUCKET THRESHOLD ACROSS CATEGORIES

We compute category-specific thresholds using a modified version of Equation (8):

$$\delta^\star_{B_T} = \mathrm{avg}\Big(\big\{\delta_t(c_i^j, c_R^j) \,\big|\, t \in [T, t^\star],\ i \in \{1, \ldots, |\mathbf{C}^j| - 1\},\ \mathbf{C}^j \in \mathbb{C}_T\big\}\Big), \tag{20}$$

where $\mathbb{C}_T$ denotes the set of prompt sequences belonging to category $T$. The computed thresholds for each model and category are summarized in Table 5. For implementation, we round each value to the nearest multiple of 5 with minor alternation to obtain better results. The reported thresholds are those used in our main results in Table 1.

Table 5: **Category-specific bucket thresholds computed for each model.**

| Methods | Single Object | | | | | Multiple Objects | | |
|---------|----------|-------|---------|--------|---------|--------|----------|---------|
| | Property | Shape | Texture | Action | Complex | Concat | Relation | Complex |
| SDXL | $1 \times 10^{-2}$ | $2.5 \times 10^{-3}$ | $3.5 \times 10^{-3}$ | $3.5 \times 10^{-3}$ | $3.5 \times 10^{-3}$ | $3.5 \times 10^{-3}$ | $3.5 \times 10^{-3}$ | $3 \times 10^{-3}$ |
| SD3 | $2.5 \times 10^{-4}$ | $1 \times 10^{-4}$ | $2.5 \times 10^{-4}$ | $1.5 \times 10^{-4}$ | $2 \times 10^{-4}$ | $2 \times 10^{-4}$ | $2 \times 10^{-4}$ | $2 \times 10^{-4}$ |
| Flux | $1 \times 10^{-4}$ | $1 \times 10^{-4}$ | $1 \times 10^{-4}$ | $1 \times 10^{-4}$ | $1 \times 10^{-4}$ | $1 \times 10^{-4}$ | $1 \times 10^{-4}$ | $1 \times 10^{-4}$ |
| Sana | $2.5 \times 10^{-4}$ | $1.5 \times 10^{-4}$ | $3 \times 10^{-4}$ | $1.5 \times 10^{-4}$ | $2.5 \times 10^{-4}$ | $3 \times 10^{-4}$ | $2.5 \times 10^{-4}$ | $3 \times 10^{-4}$ |

## E.3  PER-PROMPT BUCKET THRESHOLD

To eliminate the need for predefined concept categories and better match practical usage, we extend the bucket-threshold estimator in Equation (20) to operate at the level of individual prompt sequences. For each sequence $\mathbf{C}^j$, we compute its own threshold:

$$\delta^\star_{j,B} = \mathrm{avg}\Big(\big\{\delta_t(c_i^j, c_R^j) \,\big|\, t \in [T, t^\star],\ i \in \{1, \ldots, |\mathbf{C}^j| - 1\}\big\}\Big), \tag{21}$$

where $\delta^\star_{j,B}$ denotes the bucket threshold tailored to the $j$-th prompt sequence.

We report results for Flux and Sana in Table 6. Across both models, RAP with per-prompt thresholds (RAP-S) performs comparably to, or in some cases better than, the category-specific variant (RAP-C). Both variants consistently outperform R2F (with the only exception in the *single-complex* case on Sana), confirming that our *adaptive switching strategy* remains effective even without relying on concept labels. While category-specific thresholds may provide slightly improved stability by reducing local semantic bias within prompt groups, per-prompt thresholds offer greater flexibility

and automatically adapt to each individual sequence. Crucially, both threshold types are derived in a fully automated manner.

In summary, these results demonstrate that RAP's bucket-threshold estimation readily generalizes beyond category-level aggregation. The method applies directly to individual prompts, unseen concept types, and real-world scenarios where category labels are unavailable, further reinforcing the practicality and breadth of RAP's applicability.

Table 6: **Comparison of using per-prompt bucket strategy.** RAP-C represents using category bucket and RAP-S represents using single prompt bucket threshold. We use **bold** for the best and underline for the second best.

| Methods | Single Object | | | | | Multiple Objects | | |
|---|---|---|---|---|---|---|---|---|
| | Property | Shape | Texture | Action | Complex | Concat | Relation | Complex |
| Flux.1-dev[†] | $58.3 \pm 6.6$ | $59.8 \pm 2.0$ | $45.2 \pm 1.9$ | $53.8 \pm 3.9$ | $56.5 \pm 2.8$ | $41.7 \pm 3.7$ | $44.2 \pm 1.3$ | $56.3 \pm 2.7$ |
| + R2F[†] | $57.3 \pm 3.8$ | $55.0 \pm 3.8$ | $48.1 \pm 6.0$ | $44.4 \pm 6.0$ | $55.0 \pm 4.4$ | $39.2 \pm 3.2$ | $41.4 \pm 4.2$ | $57.3 \pm 3.1$ |
| + RAP-C | $\mathbf{67.7} \pm 7.8$ | $\underline{60.2} \pm 1.4$ | $\mathbf{52.1} \pm 5.7$ | $\underline{54.6} \pm 2.5$ | $\mathbf{59.6} \pm 2.6$ | $\mathbf{43.9} \pm 2.4$ | $\underline{46.2} \pm 0.7$ | $\mathbf{57.9} \pm 0.8$ |
| + RAP-S | $\underline{64.4} \pm 0.0$ | $\mathbf{66.0} \pm 4.8$ | $\underline{49.2} \pm 1.3$ | $\mathbf{56.2} \pm 3.2$ | $\underline{56.0} \pm 3.1$ | $\underline{42.5} \pm 4.4$ | $\mathbf{46.9} \pm 4.3$ | $57.3 \pm 2.2$ |
| Sana[†] | $61.7 \pm 0.3$ | $59.0 \pm 4.9$ | $73.1 \pm 1.7$ | $69.3 \pm 5.0$ | $69.8 \pm 3.2$ | $45.4 \pm 7.3$ | $36.0 \pm 1.3$ | $63.5 \pm 0.3$ |
| + R2F[†] | $82.1 \pm 3.6$ | $61.1 \pm 6.3$ | $73.6 \pm 1.6$ | $80.8 \pm 1.9$ | $\underline{73.2} \pm 1.1$ | $53.1 \pm 4.4$ | $45.2 \pm 0.9$ | $64.1 \pm 1.3$ |
| + RAP-C | $\mathbf{82.9} \pm 2.9$ | $\underline{65.4} \pm 3.6$ | $\underline{74.4} \pm 1.9$ | $\underline{81.0} \pm 4.7$ | $\mathbf{73.6} \pm 1.4$ | $\mathbf{54.6} \pm 2.0$ | $\mathbf{45.6} \pm 1.6$ | $\mathbf{65.0} \pm 1.7$ |
| + RAP-S | $\underline{82.5} \pm 2.3$ | $\mathbf{66.9} \pm 0.7$ | $\mathbf{74.6} \pm 2.8$ | $\mathbf{81.1} \pm 3.7$ | $70.4 \pm 3.0$ | $\underline{53.4} \pm 1.3$ | $\mathbf{45.6} \pm 1.0$ | $\underline{64.8} \pm 2.9$ |

## F  T2I-CompBench Setup

T2I-CompBench contains 1,800 prompts across six categories, but most have $|\mathbf{C}| = 1$, indicating that the LLM judged no rare concept to be present. In such cases, both R2F and our method revert to the original model setup without prompt switching or alternation. We therefore focus on prompts with $|\mathbf{C}| \geq 2$, with the detailed distribution shown in Figure 14. Since the original BLIP-based evaluation can produce misleading results, *i.e.*, images with better text alignment may receive lower scores, we follow the RareBench setup and adopt `GPT-4o` for evaluation. To control evaluation cost with the

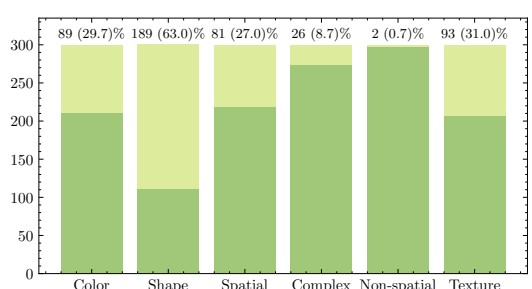

Figure 14: **Distribution of T2I-CompBench Prompts.** The $x$-axis denotes categories and the $y$-axis the number of prompts. ▨ indicates $|\mathbf{C}| \geq 2$, and ▨ indicates $|\mathbf{C}| = 1$.

API, we randomly select up to 60 prompts per category, yielding a final set of 268 prompts.

## G  Additional Results

### G.1  Aesthetic Score

We report the aesthetic scores on RareBench (Park et al., 2025) using the LAION-Aesthetics Predictor V2.5 model[6] in Table 7. Our method achieves higher scores across several categories and obtains the highest average compared to R2F. These results indicate that our approach not only produces more faithful text-to-image generations but also yields outputs that are more visually appealing.

### G.2  Additional Visualizations

We provide additional visualization comparisons in Figure 15. The first four rows present model-specific results on SDXL and SD3 with prompts from RareBench, while the last six rows show further comparisons on both RareBench and T2I-CompBench.

---

[6]`https://github.com/discus0434/aesthetic-predictor-v2-5`

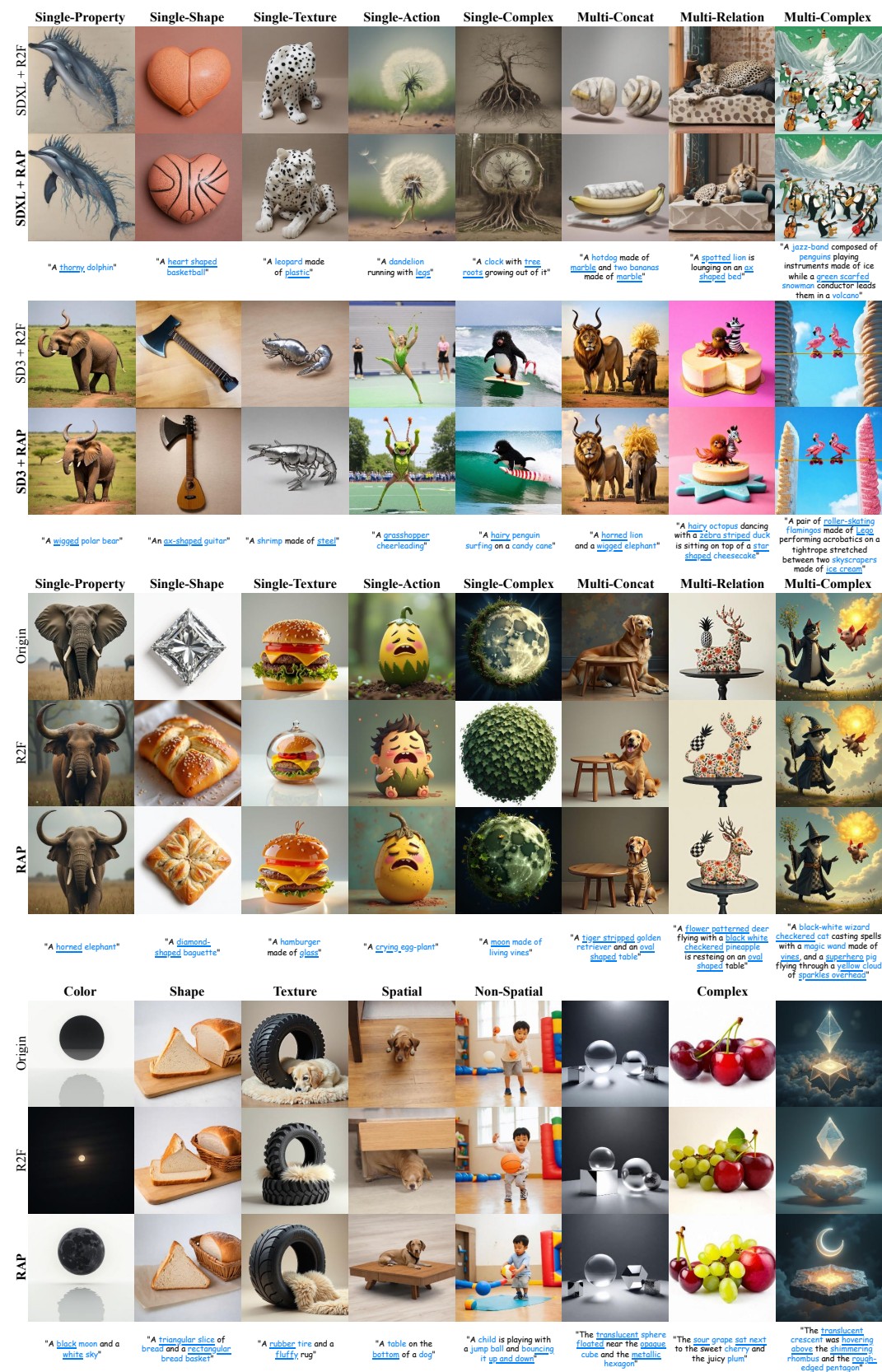

Figure 15: **Visualization Comparison**. The first four rows are model-specific from RareBench. The last six rows show more comparison from RareBench and T2I-CompBench

Table 7: **Aesthetic Score on RareBench.** Results are reported on the same set of images as in Table 1.

| Methods | | Single Object | | | | | Multiple Objects | | | Avg. |
|---|---|---|---|---|---|---|---|---|---|---|
| | | Property | Shape | Texture | Action | Complex | Concat | Relation | Complex | |
| SDXL | + R2F | $5.40 \pm 0.20$ | $5.44 \pm 0.12$ | $5.45 \pm 0.13$ | $5.34 \pm 0.04$ | $5.69 \pm 0.10$ | $5.35 \pm 0.11$ | $\mathbf{5.46} \pm 0.08$ | $5.94 \pm 0.08$ | 5.51 |
| | + RAP | $\mathbf{5.43} \pm 0.28$ | $\mathbf{5.50} \pm 0.08$ | $\mathbf{5.51} \pm 0.10$ | $\mathbf{5.45} \pm 0.04$ | $\mathbf{5.71} \pm 0.10$ | $\mathbf{5.37} \pm 0.14$ | $5.44 \pm 0.09$ | $\mathbf{5.97} \pm 0.03$ | **5.55** |
| SD3 | + R2F | $5.75 \pm 0.16$ | $5.23 \pm 0.15$ | $5.41 \pm 0.15$ | $\mathbf{5.11} \pm 0.07$ | $5.39 \pm 0.06$ | $5.23 \pm 0.10$ | $5.29 \pm 0.04$ | $\mathbf{5.51} \pm 0.01$ | 5.36 |
| | + RAP | $\mathbf{5.81} \pm 0.13$ | $\mathbf{5.26} \pm 0.26$ | $\mathbf{5.47} \pm 0.12$ | $5.05 \pm 0.02$ | $5.39 \pm 0.04$ | $\mathbf{5.25} \pm 0.15$ | $\mathbf{5.35} \pm 0.04$ | $5.49 \pm 0.09$ | **5.38** |
| Flux | + R2F | $5.91 \pm 0.05$ | $5.78 \pm 0.08$ | $5.85 \pm 0.02$ | $5.73 \pm 0.10$ | $5.87 \pm 0.10$ | $5.63 \pm 0.05$ | $5.57 \pm 0.06$ | $5.94 \pm 0.04$ | 5.78 |
| | + RAP | $\mathbf{5.98} \pm 0.05$ | $\mathbf{5.81} \pm 0.06$ | $\mathbf{5.88} \pm 0.12$ | $\mathbf{5.77} \pm 0.08$ | $\mathbf{5.96} \pm 0.03$ | $\mathbf{5.68} \pm 0.05$ | $\mathbf{5.60} \pm 0.10$ | $\mathbf{5.99} \pm 0.02$ | **5.83** |
| Sana | + R2F | $5.71 \pm 0.04$ | $\mathbf{5.43} \pm 0.11$ | $5.51 \pm 0.06$ | $\mathbf{5.48} \pm 0.12$ | $5.83 \pm 0.04$ | $\mathbf{5.57} \pm 0.06$ | $5.58 \pm 0.05$ | $5.74 \pm 0.10$ | 5.60 |
| | + RAP | $5.71 \pm 0.06$ | $5.39 \pm 0.15$ | $\mathbf{5.54} \pm 0.12$ | $5.46 \pm 0.08$ | $\mathbf{5.85} \pm 0.04$ | $5.53 \pm 0.06$ | $\mathbf{5.62} \pm 0.07$ | $\mathbf{5.75} \pm 0.08$ | **5.61** |

## H  EXAMPLE OF PROMPT SEQUENCE IN RAREBENCH

We provide an illustrative example from RareBench in Table 8. Details for the system prompt and the postprocessing can be found in Park et al. (2025). Given a rare prompt such as "`a hairy dolphin and two wrinkled sharks`," the LLM automatically estimates its difficulty and generates both the prompt sequence $\mathbf{C} \triangleq \{c_1, c_2, \ldots, c_n\}$ with $c_n \equiv c_R$ and the corresponding visual level $\mathbf{V} \triangleq \{v_1, v_2, \ldots, v_{n-1}\}$. In this example, the most frequent prompt replaces the specific terms "dolphin" and "shark" with the more generic "animal". Thus, in the initial steps, the model first captures the attributes "hairy" and "wrinkled" at a coarse semantic level. As denoising proceeds, the subject is gradually refined back to its original meaning (*i.e.*, "dolphin" and "sharks") according to the visual schedule.

Notably, the visual levels $v_i$ can be rescaled to $v_i'$ if the total inference timestep $T'$ differs from the original schedule $T$:

$$v_i' = T' \cdot \left( \frac{v_i}{T} \right).$$

For example, if the visual levels are defined under $T = 1000$ but inference uses only $T' = 25$ timesteps, then $v_1' = 10$ and $v_2' = 5$.

Table 8: **Example of the frequent-to-rare prompt sequence.**

| Prompt ($c_i$) | Visual Level ($v_i$) |
|---|---|
| $c_1 = $ a hairy animal and two wrinkled animals | $v_1 = 400$ |
| $c_2 = $ a hairy dolphin and two wrinkled animals | $v_2 = 200$ |
| $c_3 = $ a hairy dolphin and two wrinkled sharks | - |

## I  ILLUSTRATION OF PROMPT STAGE AND SCORE DIFFERENCE

We adopt an example from the Flux "Multi-Complex" category in Figure 15. The full prompt sequence with $|\mathbf{C}| = 4$ is listed in Table 9, and the corresponding results are shown in Figure 16. Different colors (blue, green, and orange) indicate the active prompt at each stage. We plot the score difference $\delta_t$ only up to $c_3$. As discussed earlier, the score difference starts small and gradually increases; with our adaptive prompt switching, this growth is controlled, keeping the trajectory closer to the rare-prompt score.

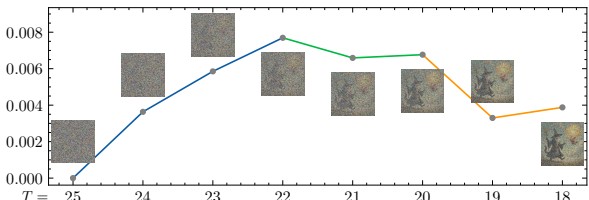

Figure 16: **Prompt stages and score differences.** The $x$-axis denotes the timestep, and the $y$-axis denotes the score difference $\delta_t$.

Table 9: **Prompt sequence for the illustration of prompt stage and score difference.**

| Prompt ($c_i$) |
| --- |
| $c_1$ = a black-white wizard checkered animal casting spells with a stick made of vines, and an animal flying through a yellow cloud of sparkles overhead |
| $c_2$ = a black-white wizard checkered animal casting spells with a stick made of vines, and a superhero pig flying through a yellow cloud of sparkles overhead |
| $c_3$ = a black-white wizard checkered cat casting spells with a stick made of vines, and a superhero pig flying through a yellow cloud of sparkles overhead |
| $c_4$ = a black-white wizard checkered cat casting spells with a magic wand made of vines, and a superhero pig flying through a yellow cloud of sparkles overhead |

## J    DETAILS OF USER STUDY

For each text prompt, two images generated by AI according to the instruction will be displayed side by side. Read the description and select the image — **Left** or **Right** — that best matches it.

- If neither half is perfect, pick the one matching **more aspects**.
- If still tied, choose the one with **better aesthetics** (composition, clarity, color, appeal).

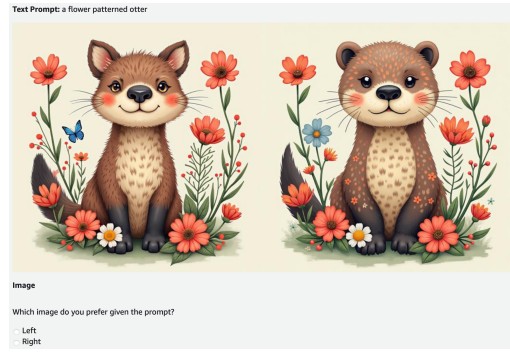

Figure 17: **User Study Instructions**.

Figure 18: **Screenshot of the user study**.

To evaluate the perceptual quality of our method, we conduct a user study involving **50** participants with Amazon MTurk[7]. All participants are anonymous to the authors, and no personal information is collected during the process. Each participant is asked to compare 64 image pairs, where each pair presents outputs from our method and a baseline model. Specifically, each participant evaluates 8 prompts per baseline model, across 8 models in total. To ensure fairness and reduce bias, the position of the images (left or right) is randomly assigned for each comparison, and no discernible pattern is introduced. Participants are instructed to select the image they prefer based on two criteria: **(i)** how well the image aligns with the given text prompt (text-to-image alignment), and **(ii)** the overall visual quality of the image. For each comparison, users are given only two options: "left", and "right". The final win rate is computed as the percentage of comparisons in which our method is preferred over the baseline. We provide the exact instruction shown to users in Figure 17 and a screenshot of the user interface in Figure 18.

**Remark.**   We carefully curated all prompts and generated images to ensure that no sensitive, offensive, or potentially distressing content was presented to participants. All comparisons involve neutral, creative visual concepts to minimize any risk of discomfort or bias during the study.

## K    LIMITATION

While our method improves rare concept generation, it is still constrained by the capabilities and training distribution of the underlying diffusion model. In cases where the rare concept lies far outside the model's learned domain, even our adaptive prompt switching can fail to yield plausible outputs. This can be found in the Figure 19. A possible extension is to incorporate step-by-step editing or intermediate supervision to gradually introduce challenging attributes, making the rare concept more feasible for the model to synthesize.

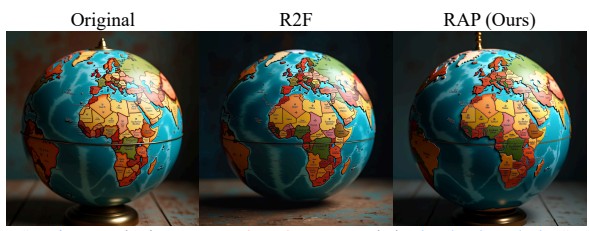

"A globe with the ocean colored brown and the land colored blue"

Figure 19: **Illustration of failure cases**.

---

[7] https://www.mturk.com

## L    USE OF LLMS

We use an LLM in two ways. **(i)** First, we use an off-the-shelf LLM solely to polish the manuscript, improving grammar, clarity, and style, and we manually verify all edits to ensure no hallucination contents. The LLM does not generate technical content, methods, or results. **(ii)** Second, we use an LLM to compute a text-to-alignment score that assesses how well model outputs align with the specified textual criteria, following previous work.

