# OpenReview forum: "Score Replacement with Bounded Deviation for Rare Prompt Generation"
_ICLR.cc/2026/Conference — Submitted to ICLR 2026_

### Official Review · Reviewer_eyNK · 2025-10-23

**Soundness:** 3
**Presentation:** 3
**Contribution:** 3
**Rating:** 4
**Confidence:** 4

**Summary:**

This paper focuses on image generation under rare concept descriptions. Unlike previous methods that adopt a fixed prompt-switching time, the authors propose an adaptive prompt-switching strategy. Specifically, the method pre-calculates a threshold to control when the prompt should switch. Experimental results demonstrate strong performance and good generalization across various models.

**Strengths:**

1. This paper first provides a mathematical analysis of semantic drift (or semantic difference) that occurs during prompt switching.

2. Based on the mathematical analysis, this paper further proposes an adaptive prompt-switching rule that generalizes well across various generation models.

3. The experiment results show the great performance and robustness.

**Weaknesses:**

1. The paper still relies on prompt switching to address the rare concept generation problem, and the novelty of the proposed approach is therefore limited.

2. The bucket threshold needs to be pre-calculated, which is time-consuming.

3. The bucket threshold is calculated from the dataset. Therefore, it is unclear whether the pre-calculated threshold can generalize to rare concepts that are not present in the dataset.

4. It is still complex to calculate the threshold under various settings, as stated in the paper: “Finally, since semantic categories (e.g., shape and texture) often induce distinct thresholds…”. This remains a practical problem in real-world scenarios.

**Questions:**

Do we need to pre-calculate different thresholds for each model when using the same dataset?

---

> ### Author Response · Authors · 2025-11-17
> **Response to Reviewer eyNK**
>
> We thank the reviewer for their thoughtful feedback. Below, we address each weakness and question point-by-point.
>
> > **W1:** The paper still relies on prompt switching to address the rare concept generation problem, and the novelty of the proposed approach is therefore limited.
>
>
> We thank the reviewer for raising this concern. While our work is indeed built on the prompt switching paradigm introduced by R2F, we believe it provides both conceptual and technical advances beyond simply "making R2F adaptive".
>
> 1. **New formulation: prompt switching as score replacement with a deviation bound**
> Prior switching methods, including R2F, operate as input-level heuristics that choose fixed switching times from an **LLM schedule**. In contrast, we reformulate prompt switching as score replacement along the probability flow ODE. Concretely, Section 3.1 shows that, under the Gaussian probability path, the denoising dynamics can be expressed purely in terms of the conditional score $\nabla_x \log p(x_t|c)$, and that replacing the rare score by a frequent score induces a concept trajectory that approximates the rare only trajectory.
> 2. **Principled adaptive controller derived from the bound, not a heuristic schedule**
> Building on this, we derive an explicit upper bound on the final latent deviation between a proxy guided trajectory and the rare only trajectory (Equation 5). This transforms prompt switching from a heuristic into a theoretically grounded approach. To the best of our knowledge, this is the first work that connects prompt switching to a formal trajectory deviation bound in the score space, turning a heuristic scheduling idea into a controlled process with an explicit error budget.
>
> In summary, the novelty of RAP lies in (i) the score replacement formulation with an explicit trajectory deviation bound, and (ii) the adaptive bucketed controller that is derived from this bound and shown to be effective across multiple modern backbones and benchmarks.
>
>
> > **W2/Q1**: The bucket threshold needs to be pre-calculated, which is time-consuming. / Do we need to pre-calculate different thresholds for each model when using the same dataset?
>
> The time required to compute the bucket threshold is modest and, importantly, can be reused. For example, on Flux, estimating the category-level threshold takes roughly two minutes. Once computed, this threshold can be applied across all experiments using the same model and dataset without recalculation, since it depends only on the model’s noise schedule and the score statistics of the dataset, both of which remain stable across runs.
>
> Although this one-time computation introduces a small overhead, it is far more efficient than latent-optimization–based methods. In latent optimization, each individual image requires more than 10× the inference time and consumes over 2× the GPU memory (approximately 85.6 GB for Flux). In contrast, RAP requires only forward passes, incurs negligible additional memory over the base model (approximately 33.9 GB), and maintains the same inference pipeline.

---

> > ### Author Response · Authors · 2025-11-17
> > **Response to Reviewer eyNK**
> >
> > > **W3/W4**: The bucket threshold is calculated from the dataset. Therefore, it is unclear whether the pre-calculated threshold can generalize to rare concepts that are not present in the dataset. / It is still complex to calculate the threshold under various settings, as stated in the paper: 'Finally, since semantic categories often induce distinct thresholds...'. This remains a practical problem in real-world scenarios.
> >
> >
> > We would like to clarify that RAP does **not fundamentally rely** on category-specific thresholds. Equation (8) can be directly applied to any given prompt sequence $\mathbf{C}^j$ to estimate its own bucket threshold $\delta_{j,B}^{\star}$ as:
> >
> > $$
> > \delta\_{j,B}^{\star}
> >     = \text{avg} (
> >         \\{
> >             \delta\_t(c^j\_i, c^j\_R)
> >             \,\big|\;
> >             t \in [T, t^\star],
> >             i \in \{1, \dots, |\mathbf{C}^j|-1\}
> >         \\}),
> > $$
> >
> > which allows threshold estimation **without knowing or specifying any semantic category**. In other words, users can compute $\delta_{j,B}^{\star}$ directly from the given prompt itself, making RAP applicable even when category information is unavailable or ambiguous.
> >
> > To better justify this, we have conducted new experiments on **Flux** and **Sana** by comparing:
> > - **RAP-C** (category-specific threshold) and
> > - **RAP-S** (single-threshold per prompt sequence).
> >
> > $\textbf{Bold}$ values indicate the best and $\underline{\text{underline}}$ values indicate the second best.
> >
> > | Approach     | Single-Property    | Single-Shape       | Single-Texture     | Single-Action      | Single-Complex     | Multi-Concat       | Multi-Relation     | Multi-Complex      |
> > | ------------ | ------------------ | ------------------ | ------------------ | ------------------ | ------------------ | ------------------ | ------------------ | ------------------ |
> > | Flux + R2F   | $57.3$             | $55.0$             | $48.1$             | $44.4$             | $55.0$             | $39.2$             | $41.5$             | $57.3$             |
> > | Flux + RAP-C | $\textbf{67.7}$    | $\underline{60.2}$ | $\textbf{52.1}$    | $\underline{54.6}$ | $\textbf{59.6}$    | $\textbf{43.9}$    | $\underline{46.2}$ | $\textbf{57.9}$    |
> > | Flux + RAP-S | $\underline{64.4}$ | $\textbf{66.0}$    | $\underline{49.2}$ | $\textbf{56.2}$    | $\underline{56.0}$ | $\underline{42.5}$ | $\textbf{46.9}$    | $57.3$             |
> > |              |                    |                    |                    |                    |                    |                    |                    |                    |
> > | Sana + R2F   | $82.1$             | $61.0$             | $73.5$             | $80.8$             | $\underline{73.1}$ | $53.1$             | $45.2$             | $64.2$             |
> > | Sana + RAP-C | $\textbf{82.9}$    | $\underline{65.4}$ | $\textbf{74.4}$    | $\underline{81.0}$ | $\textbf{73.6}$    | $\textbf{54.6}$    | $\textbf{45.6}$    | $\textbf{65.0}$    |
> > | Sana + RAP-S | $\underline{82.5}$ | $\textbf{66.9}$    | $\underline{74.6}$ | $\textbf{81.1}$    | $70.4$             | $\underline{53.3}$ | $\textbf{45.6}$    | $\underline{64.8}$ |
> >
> > The results show that RAP-S often perform comparably to, or even better than, category-specific thresholds (RAP-C). Both consistently surpass R2F, confirming that our *adaptive switching strategy* remains effective regardless of whether category information is used. Moreover, we observe that category-specific thresholds can sometimes yield slightly higher stability by mitigating local bias within similar prompt groups, while per-prompt thresholds adapt more flexibly to individual cases. In summary, these new results demonstrate that RAP’s bucket-threshold estimation generalizes beyond category-level analysis. RAP can be directly applied to individual prompts, unseen categories, or real-world settings where category labels are unavailable, reinforcing its practicality and broad applicability. Accordingly, we have added a new section to our revised paper that discusses the per-prompt bucket threshold in Appendix E.3.

---

### Official Review · Reviewer_jdmt · 2025-10-31

**Soundness:** 3
**Presentation:** 3
**Contribution:** 1
**Rating:** 6
**Confidence:** 2

**Summary:**

This paper tackles rare-prompt text-to-image generation by improving the framework, which utilizes prompt switching. Specifically, the authors reframe "prompt switching" as score replacement: early in sampling, they guide the denoiser with a semantically related, frequent proxy prompt whose score closely matches the rare prompt’s score, then switch to the rare prompt once that proxy starts to drift. They measure the score difference over timesteps and introduce RAP, which is equipped with a heuristic rule with per-segment budget that bounds the accumulated deviation from the rare-prompt trajectory. When the budget is exceeded, they switch to the rare prompt. The bucket threshold is estimated from an early stable regime of score differences, and a decay factor allocates larger budgets to prompts closer to the rare target, making the schedule model-agnostic and prompt-aware.

**Strengths:**

- The paper is well written and systematically analyzes the limitations of existing rare-prompt switching pipelines. The proposed controller addresses those issues cleanly and yields consistent improvements across models.

**Weaknesses:**

- The method's effectiveness may lean on several choices and hyperparameters that feel more engineered than principled. The bucket threshold is estimated from an empirically detected stable region, which can depend on the backbone generative model. Category- and model-specific thresholds further improve results but also signal configuration fragility: gains partly come from per-benchmark tailoring rather than a single robust rule.

- Minor: While the paper substantially strengthens R2F with a principled, adaptive controller, the contribution is still anchored to the R2F formulation and evaluation protocol. As a result, the impact feels scoped to a niche use case (rare-prompt switching) rather than a broadly applicable T2I strategy.

**Questions:**

N/A

---

> ### Author Response · Authors · 2025-11-17
> **Response to Reviewer jdmt**
>
> We thank the reviewer for their thoughtful feedback. Below, we address each weakness point-by-point.
>
> > **W1**: The bucket threshold is estimated from an empirically detected stable region, which can depend on the backbone generative model. Category- and model-specific thresholds further improve results but also signal configuration fragility: gains partly come from per-benchmark tailoring rather than a single robust rule.
>
>
> We would like to clarify that RAP does **not fundamentally rely** on category-specific thresholds. Equation (8) can be directly applied to any given prompt sequence $\mathbf{C}^j$ to estimate its own bucket threshold $\delta_{j,B}^{\star}$ as:
>
> $$
> \delta\_{j,B}^{\star}
>     = \text{avg} (
>         \\{
>             \delta\_t(c^j\_i, c^j\_R)
>             \,\big|\;
>             t \in [T, t^\star],
>             i \in \{1, \dots, |\mathbf{C}^j|-1\}
>         \\}),
> $$
>
> which allows threshold estimation **without knowing or specifying any semantic category**. In other words, users can compute $\delta_{j,B}^{\star}$ directly from the given prompt itself, making RAP applicable even when category information is unavailable or ambiguous.
>
> To better justify this, we have conducted new experiments on **Flux** and **Sana** by comparing:
> - **RAP-C** (category-specific threshold) and
> - **RAP-S** (single-threshold per prompt sequence).
>
> $\textbf{Bold}$ values indicate the best and $\underline{\text{underline}}$ values indicate the second best.
>
> |Approach|Single-Property|Single-Shape|Single-Texture|Single-Action|Single-Complex|Multi-Concat|Multi-Relation|Multi-Complex|
> |------------|------------------|------------------|------------------|------------------|------------------|------------------|------------------|------------------|
> |Flux+R2F|$57.3$|$55.0$|$48.1$|$44.4$|$55.0$|$39.2$|$41.5$|$57.3$|
> |Flux+RAP-C|$\textbf{67.7}$|$\underline{60.2}$|$\textbf{52.1}$|$\underline{54.6}$|$\textbf{59.6}$|$\textbf{43.9}$|$\underline{46.2}$|$\textbf{57.9}$|
> |Flux+RAP-S|$\underline{64.4}$|$\textbf{66.0}$|$\underline{49.2}$|$\textbf{56.2}$|$\underline{56.0}$|$\underline{42.5}$|$\textbf{46.9}$|$57.3$|
> ||||||||||
> |Sana+R2F|$82.1$|$61.0$|$73.5$|$80.8$|$\underline{73.1}$|$53.1$|$45.2$|$64.2$|
> |Sana+RAP-C|$\textbf{82.9}$|$\underline{65.4}$|$\textbf{74.4}$|$\underline{81.0}$|$\textbf{73.6}$|$\textbf{54.6}$|$\textbf{45.6}$|$\textbf{65.0}$|
> |Sana+RAP-S|$\underline{82.5}$|$\textbf{66.9}$|$\underline{74.6}$|$\textbf{81.1}$|$70.4$|$\underline{53.3}$|$\textbf{45.6}$|$\underline{64.8}$|
>
> The results show that RAP-S often perform comparably to, or even better than, category-specific thresholds (RAP-C). Both consistently surpass R2F, confirming that our *adaptive switching strategy* remains effective regardless of whether category information is used. Moreover, we observe that category-specific thresholds can sometimes yield slightly higher stability by mitigating local bias within similar prompt groups, while per-prompt thresholds adapt more flexibly to individual cases. In summary, these new results demonstrate that RAP’s bucket-threshold estimation generalizes beyond category-level analysis and RAP can be directly applied to individual prompts, unseen categories, or real-world settings where category labels are unavailable, reinforcing its practicality and broad applicability. Accordingly, we have added a new section to our revised paper that discusses the per-prompt bucket threshold in Appendix E.3.
>
> > **W2**: While the paper substantially strengthens R2F with a principled, adaptive controller, the contribution is still anchored to the R2F formulation and evaluation protocol. As a result, the impact feels scoped to a niche use case (rare-prompt switching) rather than a broadly applicable T2I strategy.
>
> While RAP is designed to address rare-prompt generation, we would like to emphasize that its adaptive score-control mechanism generalizes naturally to standard text-to-image (T2I) settings. To verify this, we conducted additional experiments on **T2I-CompBench**, which primarily contains *normal, non-rare* prompts. As shown in Table 2 of the main paper, RAP mostly matches or outperforms the base diffusion models and R2F across all backbones (SDXL, SD3, Flux, and Sana).
>
> It is worth noting that R2F struggles on T2I-CompBench because its fixed switching schedule may over- or under-emphasize proxy prompts (e.g., the *Color* category in SDXL or the *Complex* category in Flux). In contrast, RAP’s bounded-deviation rule dynamically adjusts switching timing based on the model’s internal score behavior, allowing it to maintain stability and accuracy even when no clear rare concept is present. This demonstrates that RAP is not limited to a niche rare-prompt scenario but instead provides a general adaptive prompt-control framework applicable to a broad range of T2I tasks.

---

> > ### Comment · Reviewer_jdmt · 2025-11-27
> >
> > I sincerely appreciate the authors' rebuttal, which addresses most of my concerns. I decide to maintain my positive score.

---

> > > ### Author Response · Authors · 2025-11-27
> > > **Official Comment by Authors**
> > >
> > > We appreciate the reviewer again for the helpful feedback and for your positive view of our work.

---

### Official Review · Reviewer_Cm9P · 2025-11-01

**Soundness:** 3
**Presentation:** 4
**Contribution:** 2
**Rating:** 6
**Confidence:** 4

**Summary:**

This paper addresses an interesting problem, diffusion models often struggle on text-to-image generation when guided by rare prompts.
To tackle this, it proposes RAP (Rare-to-Adapt via score replacement).
Specifically, early denoising uses the score of a semantically related, frequent proxy prompt to stabilize generation, then adaptively switches to the rare prompt once an estimated score-deviation budget is exceeded.
The paper further derives a theoretical bound linking final-sample deviation to cumulative score differences along the trajectory, and instantiates a practical bucketed switching rule whose budget decays with the text similarity between the proxy and rare prompts.
Extensive experiments across multiple diffusion backbones show consistent gains over prompt-switching baselines (e.g., R2F) on rare-prompt benchmarks and in human-preference studies.

**Strengths:**

1. It reframes prompt switching as a score-aware control problem, introducing an adaptive score-replacement trigger instead of brittle, fixed schedules.
2. A clear theoretical bound ties final-sample deviation to accumulated score differences, giving principled guidance for when to switch prompts.
3. The method is practical and backbone-agnostic, requiring no architectural changes or extra training, and works across SDXL/SD3/Flux/Sana. Empirically, it consistently outperforms R2F on rare-prompt benchmarks and is supported by human preference studies.

Overall, it delivers a significant, deployable improvement for rare-prompt fidelity while keeping implementation complexity low.

**Weaknesses:**

1. Score-difference tracking and similarity-based budgeting introduce computational overhead. However, the paper does not quantify this cost—e.g., wall-clock time, GPU-hours, and memory. Therefore, comparing these metrics against prompt-switching baselines is necessary.

2. In light of the possible brittleness of early-regime budgeting across models and prompts, please report robustness evidence or adaptation strategies spanning backbone changes, prompt diversity, and noise-schedule shifts.

3. The main paper acknowledges dependence on proxy-prompt quality and the chosen similarity metric, but sensitivity analyses for proxy/encoder choices are missing, and there is no automatic fallback for low-similarity scenarios.

**Questions:**

1. How are proxy prompts chosen when multiple candidates exist or similarity is low? How sensitive is RAP to the choice of text encoder (CLIP vs. T5) and similarity metric?

2. When does RAP hurt fidelity (e.g., very low similarity, proxy semantically drifts)? Can the method detect such cases on the fly?

3. What is the overhead of score-difference tracking and similarity computation at inference time?

4. Are hyperparameters transferable across SDXL/SD3/Flux/Sana, or re-tuned per model?

5. How interpretable is the switching timeline to practitioners?

---

> ### Author Response · Authors · 2025-11-17
> **Response to Reviewer Cm9P**
>
> We thank the reviewer for their thoughtful feedback. Below, we address each weakness and question point-by-point.
>
> > **W1/Q3**: Score-difference tracking and similarity-based budgeting introduce computational overhead. However, the paper does not quantify this cost—e.g., wall-clock time, GPU-hours, and memory.
>
>
> Thank you for this important suggestion. We have now added a quantitative analysis of the computational cost, reported below. All numbers are measured on a single H100 GPU and averaged over 3 runs per configuration. We evaluate RAP with different prompt sequence lengths $|\mathbf{C}| = \{2, 5\}$.
>
>
> | Model | Time (s) | Peak Mem (GB) | Overhead |
> |-------|----------|---------------|----------|
> | SDXL | 2.664 | 8.98 | — |
> | + RAP ($\|\mathbf{C}\|=2$) | 2.978 | 8.98 | +11.8% |
> | + RAP ($\|\mathbf{C}\|=5$) | 3.153 | 8.98 | +18.3% |
> | SD3 | 2.336 | 16.91 | — |
> | + RAP ($\|\mathbf{C}\|=2$) | 2.608 | 16.91 | +11.7% |
> | + RAP ($\|\mathbf{C}\|=5$) | 2.802 | 16.93 | +20.0% |
> | Flux | 8.123 | 33.85 | — |
> | + RAP ($\|\mathbf{C}\|=2$) | 10.115 | 33.85 | +24.5% |
> | + RAP ($\|\mathbf{C}\|=5$) | 10.623 | 33.86 | +30.8% |
> | Sana | 1.040 | 10.18 | — |
> | + RAP ($\|\mathbf{C}\|=2$) | 1.201 | 10.18 | +15.5% |
> | + RAP ($\|\mathbf{C}\|=5$) | 1.441 | 10.19 | +38.6% |
>
>
> The additional cost mainly comes from two sources:
> (1) computing denoising scores for both the current proxy prompt and the rare target prompt, and
> (2) evaluating the score difference $\delta_t$ at each timestep. Several design choices keep this overhead moderate:
>
> - **Early termination**:  Once the final rare prompt is reached, RAP automatically falls back to the original single-prompt mode, eliminating the need for dual-prompt computation in later steps. Consequently, the computational overhead does not double despite computing two score functions.
> - **Batched computation**: We compute latents for current and rare prompts in batch-manner, avoiding sequential overhead.
> - **Memory efficiency**: Additional memory usage (extra latent activations and prompt embeddings) is negligible, as shown by the consistent peak memory across all configurations.
>
> Overall, RAP introduces a modest runtime overhead (about 10–30 percent depending on $|\mathbf{C}|$ and the backbones) while leaving memory consumption essentially unchanged.
>
> Accordingly, we have added Section 4.6 to discuss the runtime and memory.

---

> > ### Author Response · Authors · 2025-11-17
> > **Response to Reviewer Cm9P**
> >
> > > **W2**: In light of the possible brittleness of early-regime budgeting across models and prompts, please report robustness evidence or adaptation strategies spanning backbone changes, prompt diversity, and noise-schedule shifts.
> >
> > One strategy to ensure robustness across different prompts and backbones is to estimate the bucket threshold on a per-prompt basis which does not rely on pre-defined category or prompts. This removes the need for category-level budgeting entirely. Specifically, Equation (8) can be applied directly to any prompt sequence $\mathbf{C}^j$ to compute its own threshold $\delta_{j,B}^{\star}$:
> >
> > $$
> > \delta\_{j,B}^{\star}
> >     = \text{avg} (
> >         \\{
> >             \delta\_t(c^j\_i, c^j\_R)
> >             \,\big|\;
> >             t \in [T, t^\star],
> >             i \in \{1, \dots, |\mathbf{C}^j|-1\}
> >         \\}),
> > $$
> >
> > which allows threshold estimation **without knowing or specifying any semantic category**. In other words, users can compute $\delta_{j,B}^{\star}$ directly from the given prompt itself, making RAP applicable even when category information is unavailable or ambiguous.
> >
> > To better justify this, we have conducted new experiments on **Flux** and **Sana** by comparing:
> > - **RAP-C** (category-specific threshold) and
> > - **RAP-S** (single-threshold per prompt sequence).
> >
> > **Bold** values indicate the best and $\underline{\text{underline}}$ values indicate the second best.
> >
> > |Approach|Single-Property|Single-Shape|Single-Texture|Single-Action|Single-Complex|Multi-Concat|Multi-Relation|Multi-Complex|
> > |------------|------------------|------------------|------------------|------------------|------------------|------------------|------------------|------------------|
> > |Flux+R2F|$57.3$|$55.0$|$48.1$|$44.4$|$55.0$|$39.2$|$41.5$|$57.3$|
> > |Flux+RAP-C|$\textbf{67.7}$|$\underline{60.2}$|$\textbf{52.1}$|$\underline{54.6}$|$\textbf{59.6}$|$\textbf{43.9}$|$\underline{46.2}$|$\textbf{57.9}$|
> > |Flux+RAP-S|$\underline{64.4}$|$\textbf{66.0}$|$\underline{49.2}$|$\textbf{56.2}$|$\underline{56.0}$|$\underline{42.5}$|$\textbf{46.9}$|$57.3$|
> > ||||||||||
> > |Sana+R2F|$82.1$|$61.0$|$73.5$|$80.8$|$\underline{73.1}$|$53.1$|$45.2$|$64.2$|
> > |Sana+RAP-C|$\textbf{82.9}$|$\underline{65.4}$|$\textbf{74.4}$|$\underline{81.0}$|$\textbf{73.6}$|$\textbf{54.6}$|$\textbf{45.6}$|$\textbf{65.0}$|
> > |Sana+RAP-S|$\underline{82.5}$|$\textbf{66.9}$|$\underline{74.6}$|$\textbf{81.1}$|$70.4$|$\underline{53.3}$|$\textbf{45.6}$|$\underline{64.8}$|
> >
> > These results show that RAP-S performs comparably to, and in many cases better than, RAP-C, while both substantially outperform R2F. This confirms that our adaptive switching rule remains effective even when threshold estimation is done at the per-prompt level without any category information. Overall, these experiments demonstrate that our approach to find bucket threshold provides a robust and model-agnostic mechanism for threshold estimation, which shows that RAP can adapt reliably to diverse and unseen prompts across different backbones, reinforcing its practicality and general applicability in real-world scenarios where category labels may be unavailable. Accordingly, we have added a new section to our revised paper that discusses the per-prompt bucket threshold in Appendix E.3.
> >
> > As for noise schedule, we evaluate RAP across both rectified flow–based models (e.g., SD3, Flux, Sana) and diffusion-based models (e.g., SDXL), while each using different noise schedules and show improved results. Theoretically, the noise schedule only scales the score coefficient and does not change the score itself, meaning our bounded-deviation formulation remains valid regardless of the underlying schedule.
> >
> > > **W3**: How are proxy prompts chosen when multiple candidates exist or similarity is low?
> >
> > Since prompt selection is not the main focus of our work, we directly utilize the prompt sequences provided by RareBench to align the setting with prior studies. While previous work did not explicitly verify the validity of these prompts, we conduct an additional analysis in Appendix C and confirm that the LLM-generated sequences exhibit high semantic continuity and validity. Our study primarily investigates when to perform prompt switching rather than how to select the prompts.

---

> > > ### Author Response · Authors · 2025-11-17
> > > **Response to Reviewer Cm9P**
> > >
> > > > **Q1**: How sensitive is RAP to the choice of text encoder (CLIP vs. T5) and similarity metric?
> > >
> > > We thank the reviewer for this question. For the global $\gamma$ setup described in Section 3.3, we use the T5 encoder to compute prompt similarity. We also evaluated CLIP and found that both T5 and CLIP encoders yield the same global $\gamma$ value. For the on-the-fly computation discussed in Appendix C.2, we leverage each model’s native text encoder (e.g., SDXL uses CLIP, while Sana uses Gemma) to compute prompt embeddings dynamically. This model-specific setup further improves performance compared to the global $\gamma$ configuration. This finding somewhat suggests that using the encoder native to each diffusion model is most effective, as the model has been trained to interpret prompts in that embedding space.
> > >
> > >
> > > > **Q2**: When does RAP hurt fidelity (e.g., very low similarity, proxy semantically drifts)? Can the method detect such cases on the fly?
> > >
> > > Large dissimilarities within the prompt sequence can indeed affect generation quality, as early-stage prompts may diverge semantically from later ones. However, unlike R2F, our approach enables on-the-fly detection through monitoring the score difference $\delta_t$. In normal cases, $\delta_t$ remains low and increases gradually along the denoising trajectory (as shown in Figure 2 and Appendix B). When an abnormally large $\delta_t$ is observed, it indicates semantic drift between the frequent and rare prompts. In such cases, RAP can feasibly skip the problematic proxy or fall back to the rare prompt directly.
> > >
> > > > **Q4**: Are hyperparameters transferable across SDXL/SD3/Flux/Sana, or re-tuned per model?
> > >
> > > RAP involves two hyperparameters: (1) the bucket threshold $\delta_B^{\star}$ and (2) the decay factor $\gamma$. As discussed in the paper, $\gamma$ depends only on the semantic similarity between prompts and is therefore prompt-specific but model-agnostic, allowing direct reuse across different backbones. For $\delta_B^{\star}$, although it is derived once from a small calibration set to match each model’s noise schedule, which can be automatically computed by Equation 8.
> > >
> > >
> > > > **Q5**: How interpretable is the switching timeline to practitioners?
> > >
> > > In our framework, prompt switching is reinterpreted as score replacement. This view provides a clear and interpretable rationale: switching ensures that the generative trajectory remains close to the rare-prompt path (prevent score deviation). We further visualize this process in Appendix I, showing that when a switch occurs, the score difference immediately drops and then gradually increases again as denoising continues. This behavior confirms that prompt switching can be understood as an interpretable mechanism for preventing excessive score deviation.

---

### Official Review · Reviewer_vSy7 · 2025-11-01

**Soundness:** 3
**Presentation:** 2
**Contribution:** 2
**Rating:** 4
**Confidence:** 4

**Summary:**

This paper tackles rare-concept text-to-image generation, focusing on the limitations of R2F’s fixed switching strategy between rare and frequent prompts. The authors propose an adaptive proxy-prompt scheduling method based on a score-approximation perspective, arguing that the denoising scores of rare and frequent prompts follow similar trajectories. By monitoring score displacement during diffusion, the method adaptively transitions from the frequent proxy to the rare target. Various experiments are presented to demonstrate the effectiveness of the method.

**Strengths:**

- **Adaptive scheduling.** The paper replaces R2F’s fixed, heuristic switching rule with an adaptive scheduling using the score of the rare prompt.

- **Comprehensive analyses.** It provides diverse supporting analyses, including score-trajectory visualization, cross-model comparisons, and derivations that validate the method’s design.

- **Theoretical grounding via score approximation.** The method offers a theoretical perspective that connects the adaptive switching behavior to the score approximation, improving interpretability.

**Weaknesses:**

- **Limited methodological novelty.** While the paper’s attempt to make R2F’s fixed strategy adaptive is meaningful, the method itself is not substantially novel and largely builds on existing switching ideas.

- **Optimality of switching approach.** It remains unclear whether prompt switching is an optimal formulation for rare-concept generation. Since this strategy inherently depends on finding a proxy prompt, it may be less efficient or generalizable than other rare-concept generation methods. (more on Questions)

- **Marginal improvements** Both quantitative results on T2I-Combench and qualitative findings from the user study show only marginal improvements over prior methods, suggesting limited practical benefit despite the adaptive formulation.

- **Increased computational cost.** Because the adaptive schedule requires pre-computing or estimating scores for multiple timesteps, it likely incurs higher computational overhead than R2F’s fixed switching strategy, reducing its practical efficiency.

**Questions:**

- As noted in the weaknesses, the paper does not show that prompt switching is an optimal or necessary formulation for rare-concept generation. Could you provide additional comparisons with other methods [1, 2], or theoretical reasoning that supports the optimality (or sufficiency) of the switching-based formulation?
- Derivation around line 647. It appears that line 647 is derived by recursively applying Eqs. (12) and (13). In this case, the superscript of ​
x_t  on the right-hand side of Eq. (13) should become C_(*|R). Could you provide a clear step-by-step derivation to confirm this?
- Please provide additional details about the score-displacement experiments: what prompts were used, how many samples were averaged.
- The authors mention category-specific thresholds, but such categorical thresholding appears to require predefined concept sets, which might limit the method’s general applicability. Is this understanding correct? Additionally, beyond the SANA results, are there experiments on other models showing the impact of these thresholds?

If the authors address these concerns appropriately, I would be inclined to raise my rating.

---
#Reference
[1] Unleashing the diversity of diffusion models through condition-annealed sampling.
[2] Minority-Focused Text-to-Image Generation via Prompt Optimization

---

> ### Author Response · Authors · 2025-11-17
> **Response to Reviewer vSy7**
>
> We thank the reviewer for the thoughtful feedback. Below, we address each weakness and question point-by-point.
> > **W1**: While the paper’s attempt to make R2F’s fixed strategy adaptive is meaningful, the method itself is not substantially novel and largely builds on existing switching ideas.
>
> We thank the reviewer for raising this concern. While our work is indeed built on the prompt switching paradigm introduced by R2F, we believe it provides both conceptual and technical advances beyond simply "making R2F adaptive".
> 1. **New formulation: prompt switching as score replacement with a deviation bound**
> Prior switching methods, including R2F, operate as input-level heuristics that choose fixed switching times from an LLM schedule. In contrast, we reformulate prompt switching as score replacement along the probability flow ODE. Concretely, Section 3.1 shows that, under the Gaussian probability path, the denoising dynamics can be expressed purely in terms of the conditional score $\nabla_x \log p(x_t|c)$, and that replacing the rare score by a frequent score induces a concept trajectory that approximates the rare only trajectory.
> 2. **Principled adaptive controller derived from the bound, not a heuristic schedule**
> Building on this, we derive an explicit upper bound on the final latent deviation between a proxy guided trajectory and the rare only trajectory (Equation 5). This transforms prompt switching from a heuristic into a theoretically grounded approach. To the best of our knowledge, this is the first work that connects prompt switching to a formal trajectory deviation bound in the score space, turning a heuristic scheduling idea into a controlled process with an explicit error budget.
>
> In summary, the novelty of RAP lies in (i) the score replacement formulation with an explicit trajectory deviation bound, and (ii) the adaptive bucketed controller that is derived from this bound and shown to be effective across multiple modern backbones and benchmarks.
>
> > **W2/Q1**: It remains unclear whether prompt switching is an optimal formulation for rare-concept generation... could you provide additional comparisons with other methods [1, 2], or theoretical reasoning that supports the optimality?
>
> We thank the reviewer for this insightful question. We agree that formally proving global optimality is unrealistic, and we do not claim that prompt switching is the only possible solution. Our goal is instead to (i) show that switching is competitive with other strong paradigms under the same setting, and (ii) provide a principled account of when and why switching should work from a score based perspective.
>
> As suggested, we have added comparisons with CADS [1] and MPrompt [2] under the same experimental settings and three random seeds on RareBench. Since MPrompt only supports SDXL, we conduct all comparisons on the SDXL backbone for fairness.
>
> |Model|Single-Property|Single-Shape|Single-Texture|Single-Action|Single-Complex|Multi-Concept|Multi-Relation|Multi-Complex|
> |-|-|-|-|-|-|-|-|-|
> |SDXL|45.0|53.5|59.6|47.5|53.1|29.2|25.4|38.7|
> |+CADS[1]|47.3|46.5|57.9|35.2|46.0|26.5|27.5|35.0|
> |+MPrompt[2]|47.7|53.1|58.1|51.7|50.8|22.9|22.7|35.4|
> |+R2F|66.0|62.3|59.0|50.2|58.8|32.5|24.6|39.4|
> |+RAP|**68.1**|**64.6**|**61.9**|**53.3**|**59.4**|**34.0**|**27.9**|**40.0**|
>
> These results manifest that the prompt switching family (R2F and RAP) remains very competitive even in the presence of advanced sampling and prompt optimization techniques. The proposed *RAP* further improves over R2F in all categories.
>
> It is worth noting that while it is difficult to formally prove that prompt switching is the optimal formulation from a theoretical perspective, switching is a natural formulation in our score replacement view. Specifically, Section 3.1 reformulates switching as score replacement and shows that, in the high noise regime, the scores of a rare prompt and a semantically-related frequent proxy remain very close, while they gradually diverge as noise decreases. Our deviation bound then expresses the final latent error as a weighted sum of these score differences along the trajectory. This gives a simple intuition:
> * Use the frequent proxy early, when its score is a good approximation of the rare score and it stabilizes the coarse structure;
> * Switch to the rare prompt once the accumulated score difference approaches the budget, so that fine rare details can be recovered without drifting too far from the ideal rare trajectory.
>
> In this sense, prompt switching is not an arbitrary heuristic, but a natural control mechanism that exploits early score similarity and late score specialization. While we do not claim it is theoretically optimal in a global sense, our analysis and the new comparisons with CADS and MPrompt suggest that switching is a strong and practical formulation for rare concept generation among training-free, model-agnostic methods. Accordingly, we have incorporated these comparisons into Section 4.1, Section 4.2, and Table 1.

---

> > ### Author Response · Authors · 2025-11-17
> > **Response to Reviewer vSy7**
> >
> > > **W3**: Both quantitative results on T2I-Combench and qualitative findings from the user study show only marginal improvements over prior methods, suggesting limited practical benefit despite the adaptive formulation.
> >
> > We would like to clarify that T2I-CompBench is designed for general, non-rare prompts, and thus most prompts in this benchmark pose relatively little challenge to modern diffusion models. As a result, the performance gap between strong baselines is inherently small. Even under this saturated setting, RAP still provides consistent improvements across multiple backbones.
> >
> > More importantly, unlike R2F, which can degrade performance on several categories (e.g., the *Color* category in SDXL or the *Complex* category in Flux), RAP's adaptive switching mechanism ensures that performance on normal prompts is never harmed. This demonstrates that RAP is not only competitive but also robust, avoiding the regressions introduced by heuristic schedules.
> >
> > Regarding the user study, the improvements are far from marginal: across four backbones, RAP achieves 60–70% win rates compared to both R2F and the original models. This reflects a clear and consistent human preference for our outputs in terms of alignment and visual quality, supporting the practical benefit of our adaptive approach.
> >
> >
> > > **W4**: Because the adaptive schedule requires pre-computing or estimating scores for multiple timesteps, it likely incurs higher computational overhead than R2F’s fixed switching strategy, reducing its practical efficiency.
> >
> >
> > We have now added a quantitative analysis of the computational cost, reported below. All numbers are measured on a single H100 GPU and averaged over 3 runs per configuration. We evaluate RAP with different prompt sequence lengths $|\mathbf{C}| = \{2, 5\}$.
> >
> >
> > |Model|Time(s)|PeakMem(GB)|Overhead|
> > |-------|----------|---------------|----------|
> > |SDXL|2.664|8.98|—|
> > |+RAP($\|\mathbf{C}\|=2$)|2.978|8.98|+11.8%|
> > |+RAP($\|\mathbf{C}\|=5$)|3.153|8.98|+18.3%|
> > |SD3|2.336|16.91|—|
> > |+RAP($\|\mathbf{C}\|=2$)|2.608|16.91|+11.7%|
> > |+RAP($\|\mathbf{C}\|=5$)|2.802|16.93|+20.0%|
> > |Flux|8.123|33.85|—|
> > |+RAP($\|\mathbf{C}\|=2$)|10.115|33.85|+24.5%|
> > |+RAP($\|\mathbf{C}\|=5$)|10.623|33.86|+30.8%|
> > |Sana|1.040|10.18|—|
> > |+RAP($\|\mathbf{C}\|=2$)|1.201|10.18|+15.5%|
> > |+RAP($\|\mathbf{C}\|=5$)|1.441|10.19|+38.6%|
> >
> >
> > The additional cost mainly comes from two sources:
> > (1) computing denoising scores for both the current proxy prompt and the rare target prompt, and
> > (2) evaluating the score difference $\delta_t$ at each timestep. Several design choices keep this overhead moderate:
> >
> > - **Early termination**:  Once the final rare prompt is reached, RAP automatically falls back to the original single-prompt mode, eliminating the need for dual-prompt computation in later steps. Consequently, the computational overhead does not double despite computing two score functions.
> > - **Batched computation**: We compute latents for current and rare prompts in batch-manner, avoiding sequential overhead.
> > - **Memory efficiency**: Additional memory usage (extra latent activations and prompt embeddings) is negligible, as shown by the consistent peak memory across all configurations.
> >
> > Overall, RAP introduces a modest runtime overhead (about 10–30 percent depending on $|\mathbf{C}|$ and the backbones) while leaving memory consumption essentially unchanged.
> >
> > Accordingly, we have updated the manuscript and added Section 4.6 to discuss the runtime and memory.

---

> ### Author Response · Authors · 2025-11-17
> **Response to Reviewer vSy7**
>
> > **Q2**: It appears that line 647 is derived by recursively applying Eqs. (12) and (13). The superscript of $x_t$ on the right-hand side of Eq. (13) should become $c_{*|R}$. Could you provide a clear step-by-step derivation?
>
> We thank the reviewer for pointing out the notation error. The first term of right-hand side of Equation (13) was miswritten and should indeed use the $c_{\star\mid R}$ for $\mathbf{x}_t$.
>
> Below, we provide a detailed step-by-step derivation to line 647.
>
> Using the probability flow ODE in Equation (14), together with the vector-field–to–score transformation in Equation (10), we obtain:
>
> **Concept trajectory:** The trajectory guided by the scheduled prompt sequence $c_\star$ in Equation (12) evolves as:
>
> $$
> d\mathbf{x}\_t^{c_\star}
>   = \big[ \kappa\_t \nabla\_{\mathbf{x}} \log p(\mathbf{x}\_t^{c\_\star} \mid c_\star)
>         + \eta\_t \mathbf{x}\_t^{c\_\star} \big]\, dt .
> $$
>
> **Reference trajectory:** The reference path using the *same latent* but conditioned on the rare prompt $c_R$ in Equation (13) evolves as:
> $$
> d\mathbf{x}\_t^{c_{\star\mid R}}
>   = \big[ \kappa\_t \nabla_{\mathbf{x}} \log p(\mathbf{x}\_t^{c\_\star} \mid c\_R)
>         + \eta\_t \mathbf{x}\_t^{c\_\star} \big]\, dt,
> \qquad
> \mathbf{x}\_T^{c\_{\star\mid R}} = \mathbf{x}\_T^{c\_\star}.
> $$
>
> Subtracting the reference trajectory from the concept trajectory and integrating from $T$ to $0$ gives
>
> $$
> \mathbf{x}\_0^{c\_\star} - \mathbf{x}\_0^{c\_{\star\mid R}}
>   = \int\_T^0 \ d\mathbf{x}\_t^{c\_\star}
>     - \int\_T^0 \ d\mathbf{x}\_t^{c\_{\star\mid R}}
>   = \int\_T^0 \kappa\_t
>         \big[
>             \nabla\_{\mathbf{x}} \log p(\mathbf{x}\_t^{c\_\star} \mid c\_\star(t))
>           - \nabla\_{\mathbf{x}} \log p(\mathbf{x}\_t^{c\_\star} \mid c\_R)
>         \big] \, + \eta\_t(\mathbf{x}\_t^{c\_\star} - \mathbf{x}\_t^{c\_\star}) dt .
> $$
> This expression corresponds to line 647 in Appendix A.2 (line 706 in the revised version).
>
> We have incorporated the expanded derivation, corrected the notation, and clarified the meaning of $\mathbf{x}\_t^{c\_{\star \mid R}}$ in the revised Section 3.2 and Appendix A.2.
>
>
> > **Q3:** Please provide additional details about the score-displacement experiments: what prompts were used, how many samples were averaged.
>
> The requested "score-displacement" plots correspond to our score-difference analysis $\delta_t(c_i, c_R)$. Below, we summarize the settings for all related figures.
>
> **(1) Figure 2 and Appendix B:**  We use the entire RareBench dataset, which contains 320 prompts. For each prompt sequence $\mathbf{C} = \{c_1, \dots, c_{|\mathbf{C}|}\}$, we group sequences by their length $|\mathbf{C}|$ and compute the score difference $\delta_t(c_i, c_R)$, i.e., $\delta_t(c_i, c_{|\mathbf{C}|}$), for every intermediate prompt $c_i$ relative to the rare prompt $c_R$. Thus, the plot for sequences of length $|\mathbf{C}|$ shows $|\mathbf{C}| - 1$ curves, and each curve is obtained by averaging over all RareBench sequences with that length.
>
> **(2) Figure 8:** For the direct score comparison between R2F and RAP, we compute $\delta_t$ using the current active prompt from each method and compare it to the original rare prompt. Since only one active prompt is used per method at each timestep, the figure contains a single curve per method. The values are averaged across all prompts in RareBench with $|\mathbf{C}| = 3$ or $|\mathbf{C}| = 4$, and the curve ends once all prompts reach their final stage ($c_{|\mathbf{C}|}$).
>
> **(3) Appendix I:** To better illustrate the prompt stages and score differences, we provide a concrete example using a single prompt sequence. Different colors denote different prompt stages, showing how prompt switching affects the score trajectory. The corresponding prompt texts are listed in the same appendix for clarity.
>
> Accordingly, we have added these clarifications to the revised paper in Section 3.1 (line 186-187) and Section 4.5 (line 468-472).

---

> > ### Author Response · Authors · 2025-11-17
> > **Response to Reviewer vSy7**
> >
> > > **Q4**: The authors mention category-specific thresholds, but such categorical thresholding appears to require predefined concept sets, which might limit the method's general applicability. Is this understanding correct? Additionally, beyond the SANA results, are there experiments on other models showing the impact of these thresholds?
> >
> > We thank the reviewer for this important question.
> >
> > First, we would like to clarify that RAP does **not fundamentally rely** on category-specific thresholds. Equation (8) can be directly applied to any given prompt sequence $\mathbf{C}^j$ to estimate its own bucket threshold $\delta_{j,B}^{\star}$ as:
> >
> > $$
> > \delta\_{j,B}^{\star}
> >     = \text{avg} (
> >         \\{
> >             \delta\_t(c^j\_i, c^j\_R)
> >             \,\big|\;
> >             t \in [T, t^\star],
> >             i \in \{1, \dots, |\mathbf{C}^j|-1\}
> >         \\}),
> > $$
> >
> > which allows threshold estimation **without knowing or specifying any semantic category**. In other words, users can compute $\delta_{j,B}^{\star}$ directly from the given prompt itself, making RAP applicable even when category information is unavailable or ambiguous.
> >
> > To better justify this, we have conducted new experiments on **Flux** and **Sana** by comparing:
> > - **RAP-C** (category-specific threshold) and
> > - **RAP-S** (single-threshold per prompt sequence).
> >
> > $\textbf{Bold}$ values indicate the best and $\underline{\text{underline}}$ values indicate the second best.
> >
> > | Approach     | Single-Property    | Single-Shape       | Single-Texture     | Single-Action      | Single-Complex     | Multi-Concat       | Multi-Relation     | Multi-Complex      |
> > | ------------ | ------------------ | ------------------ | ------------------ | ------------------ | ------------------ | ------------------ | ------------------ | ------------------ |
> > | Flux + R2F   | $57.3$             | $55.0$             | $48.1$             | $44.4$             | $55.0$             | $39.2$             | $41.5$             | $57.3$             |
> > | Flux + RAP-C | $\textbf{67.7}$    | $\underline{60.2}$ | $\textbf{52.1}$    | $\underline{54.6}$ | $\textbf{59.6}$    | $\textbf{43.9}$    | $\underline{46.2}$ | $\textbf{57.9}$    |
> > | Flux + RAP-S | $\underline{64.4}$ | $\textbf{66.0}$    | $\underline{49.2}$ | $\textbf{56.2}$    | $\underline{56.0}$ | $\underline{42.5}$ | $\textbf{46.9}$    | $57.3$             |
> > |              |                    |                    |                    |                    |                    |                    |                    |                    |
> > | Sana + R2F   | $82.1$             | $61.0$             | $73.5$             | $80.8$             | $\underline{73.1}$ | $53.1$             | $45.2$             | $64.2$             |
> > | Sana + RAP-C | $\textbf{82.9}$    | $\underline{65.4}$ | $\textbf{74.4}$    | $\underline{81.0}$ | $\textbf{73.6}$    | $\textbf{54.6}$    | $\textbf{45.6}$    | $\textbf{65.0}$    |
> > | Sana + RAP-S | $\underline{82.5}$ | $\textbf{66.9}$    | $\underline{74.6}$ | $\textbf{81.1}$    | $70.4$             | $\underline{53.3}$ | $\textbf{45.6}$    | $\underline{64.8}$ |
> >
> > The results show that RAP-S often perform comparably to, or even better than, category-specific thresholds (RAP-C). Both consistently surpass R2F, confirming that our *adaptive switching strategy* remains effective regardless of whether category information is used. Moreover, we observe that category-specific thresholds can sometimes yield slightly higher stability by mitigating local bias within similar prompt groups, while per-prompt thresholds adapt more flexibly to individual cases. In summary, our method can be directly applied to individual prompts, unseen categories, or real-world settings where category labels are unavailable, reinforcing its practicality and broad applicability. Accordingly, we have added a new section to our revised paper that discusses the per-prompt bucket threshold in Appendix E.3.
> >
> > Additionally, these results indicate that models beyond Sana, such as Flux, are also sensitive to the choice of bucket threshold. Since the threshold controls how closely the trajectory must adhere to the rare prompt, an inappropriate value may cause the model to drift or to switch too early. Our new experiments demonstrate that, with our proposed estimation rule, both the per-prompt and category-level thresholds serve as effective strategies, each providing substantially better performance than the heuristic schedules used in R2F.

---

> ### Comment · Reviewer_vSy7 · 2025-11-27
>
> Thank you for the detailed response. All my concerns have been fully addressed. The additional experiments provided by the authors clearly strengthen the paper and better demonstrate its contribution to the community. I will raise my rating accordingly.

---

> > ### Author Response · Authors · 2025-11-27
> > **Official Comment by Authors**
> >
> > We thank the reviewer for taking another look and for raising the score! We really appreciate the time and thoughtful feedback for making our paper better.

---

### Author Response · Authors · 2025-11-27
**Official Comment by Authors**

We would like to sincerely thank all reviewers for the very helpful and constructive feedback. Following the reviewers' comments, we have revised the paper and added several new analyses and experiments, including:

1. **New comparisons to position prompt switching vs. other paradigms.**
We added additional comparisons with CADS and MPrompt on RareBench (SDXL backbone; 3 seeds), showing that switching-based methods (R2F and RAP) remain competitive and that RAP consistently improves over R2F under the same setting.
2. **Quantified runtime + memory overhead.**
We added a wall-clock runtime and peak-memory analysis on a single H100 across SDXL, SD3, Flux, and Sana, reporting overhead for different prompt-sequence lengths and explaining why the overhead stays moderate.
3. **Flexible thresholding without category-specific tuning.**
We added new experiments on Flux and Sana comparing category-specific thresholds (RAP-C) vs. per-prompt thresholds (RAP-S), showing that RAP does not fundamentally rely on predefined categories and remains strong when thresholds are estimated directly from each prompt sequence. This further makes our method flexible when the categories or subset is unknown.
4. **Clarified derivations, notation, and experimental details.**
We corrected the noted notation issue in Equation. (13), added a step-by-step derivation to the referenced line in the appendix, and expanded the description of the score-difference ("score-displacement") experiments (prompts used, averaging protocol, and which subsets are plotted).

If time permits, we would greatly appreciate it if the reviewers could let us know whether the main concerns, especially about (i) novelty beyond heuristic switching and (ii) robustness/general applicability of the thresholding strategy, have been adequately addressed, or if there are remaining key issues we should further clarify during the discussion phase. Your feedback is very valuable for us to further improve this work. Thank you again for your time and consideration.

---

### Author Response · Authors · 2025-11-30
**Summary of Rebuttal and Revisions**

We sincerely thank all reviewers for their thoughtful and constructive feedback. In the following, we summarize how the main concerns have been addressed in our revised manuscript.
1. **Positioning and conceptual contribution of RAP**
Several reviewers (vSy7, eyNK, jdmt) asked how RAP should be positioned relative to R2F and whether it goes beyond an improved heuristic schedule. We clarify the main conceptual contributions:
    - We reframe prompt switching as **score replacement**, and show that the denoising trajectory under a scheduled prompt sequence can be analyzed in the score space.
    - We derive an **explicit deviation bound** (Equation 5) that links the final latent error to the accumulated score differences between the frequent and rare prompts.
    - Based on this bound, we design a **bounded deviation** that triggers switching when the score budget is reached, replacing heuristic schedules with a theory guided rule.

    Together, these elements clarify that RAP turns prompt switching into a controlled, score-aware process, rather than a purely heuristic tweak of R2F.
2. **Competitiveness of prompt switching vs. other paradigms**
Reviewer vSy7 asked whether switching is competitive compared to alternative approaches such as CADS and MPrompt. To address this, we added additional comparisons on RareBench. The new results, reported in Table 1 and Sections 4.1–4.2, show that switching based methods (R2F and RAP) remain very competitive against these stronger baselines and RAP consistently improves over R2F across all RareBench categories.

    We also discuss on when switching is expected to work, using the score similarity between rare and frequent prompts as intuition.
3. **Computational overhead and practicality**
Multiple reviewers (vSy7, Cm9P, eyNK) asked about the runtime and memory cost of tracking score differences and budgeting. We have added a analysis under different prompt sequence length on a single H100 across different models (Section 4.6):
    - RAP introduces roughly 10-30 percent runtime overhead, while peak memory remains essentially unchanged.
    - The overhead is kept moderate by batched computation of frequent and rare scores and by early termination once the final rare prompt is reached.

    Compared to latent optimization methods that require far higher cost, RAP preserves the original sampling pipeline and remains practical for deployment.
4. **Bucket thresholds, robustness, and encoder choice**
Reviewers (vSy7, Cm9P, jdmt, eyNK) were concerned that category specific thresholds might hurt generality, require per model tuning, or depend on the dataset. In response:
    - We clarify that RAP **does not rely on predefined categories**. Equation 8 allows estimating a **per prompt threshold** directly from a given prompt sequence.
    - New experiments on Flux and Sana compare **RAP-C** (category specific) and **RAP-S** (per prompt). RAP-S often performs comparably to or better than RAP-C, and both clearly outperform R2F. These results are reported in Appendix E.3.
    - For text encoders, we show that T5 and CLIP give similar global decay factors, while using the **native text encoder of each backbone** (for example CLIP for SDXL and Gemma for Sana) provides slightly better performance.

    These results support a robust and model agnostic thresholding strategy that can be applied to unseen prompts and backbones without relying on pre-defined categories.
5. **Scope beyond rare prompts and clarifications of details**
Reviewer jdmt raised the concern that RAP might be limited to rare prompt switching. We therefore emphasize and clarify the results on **T2I-CompBench**, which mainly contains non-rare prompts (Table 2), that RAP consistently matches or improves the base models and avoids the regressions sometimes caused by R2F’s fixed schedules. This shows that RAP serves as a general prompt switching rather than a method restricted only to rare prompts.
---
We also address several requests for clarification:

- A full step by step derivation and correct the notation issue pointed out (Appendix A.2). (vSy7)
- Detailed descriptions of the score difference experiments, including prompts, averaging protocols, and figure settings (Section 3.1 and Section 4.5). (vSy7)
- Visualizations of the switching timeline to illustrate how the score difference drops after each switch and then grows again, making the switching interpretable (Appendix I). (jdmt)

### **Follow up from reviewers**
Two reviewers explicitly confirmed that our revisions addressed their earlier concerns:

- **Reviewer vSy7** stated that all concerns were fully addressed and **raised the rating from 4 to 6**.
- **Reviewer jdmt** wrote that the rebuttal resolved most concerns and explicitly stated that they would **maintain a positive score**.

The other reviewers did not raise additional issues after the rebuttal. We thank again for all the support, organization, and invaluable feedback from reviewer and AC.

---

### Meta-Review · Area_Chair_4ynf · 2025-12-21

**Summary:**

This paper addresses the challenge of generating “rare concepts” in text-to-image diffusion models, where standard models often fail to reflect prompts that are underrepresented in training data. Building upon the “Rare-to-Frequent” (R2F) prompt switching paradigm, the authors propose RAP(Rare Concept Generation via Adaptive Prompt Switching). Unlike R2F, which relies on fixed, LLM-defined schedules, RAP interprets prompt switching through the lens of score replacement. They introduce an adaptive mechanism that monitors the score difference between a frequent proxy prompt and a rare target prompt, triggering a switch when a cumulative budget is exceeded.

During the rebuttal, the reviewers’ concerns are partially addressed:

(1)Generality and Heuristics: Reviewers vSy7 and jdmt initially questioned the fragility of the thresholding hyperparameters. While the authors introduced “RAP-S” (per-prompt thresholding) during the rebuttal to mitigate the need for category labels, this essentially replaces one heuristic with another. The method still relies on a complex, multi-step engineering pipeline involving pre-computation and runtime monitor, rather than a robust, fundamental improvement to the generative model itself.

(2) Computational Cost: Reviewers raised concerns about the overhead. The authors admitted to a 10-30% runtime increase. For a method designed to fix specific prompt failures, a ~30% latency penalty on inference is a significant drawback that undermines practical deployment.

(3) I also have an extra major concern on its practical application. In real-world applications, we do not know whether or not the prompts given by the users belong to rare concepts. This method may lead to extra possible degraded quality with inevitable latency disadvantage for general prompts. Note that the experiments of Table 2 on T2I-Bench is not comprehensive enough. Evaluation metrics, such as HPSv2 and PickScore, are more common metrics for general t2i generation.

This is a borderline submission for ICLR 2026.

Given the remaining concerns, I tend to recommend the submission for Reject.

**Reviewer Concerns:**

I think the authors did not duly addressed the core concerns .

The main concerns are still outstanding.

(1) Incremental Novelty: As Reviewer eyNK correctly identified, the core contribution remains firmly anchored in the “prompt switching” paradigm established by prior work (R2F). The theoretical framing of “score replacement” provides a formal language for the phenomenon but does not fundamentally change the underlying mechanism: it is still a schedule for swapping text conditions. This represents an incremental engineering refinement rather than the conceptual advance.

(2) Operational Friction and Complexity: Unlike standard inference techniques, this method requires a “calibration” step and dual-path score evaluation during sampling. Reviewer eyNK noted this friction. The requirement to pre-compute threshold for every prompt sequence (even with RAP-S) creates a non-trivial barrier to real-time or interactive applications, making the method less “plug-and-play” than portrayed.

(3) Computational Cost: A ~30% latency penalty on inference is a significant drawback that undermines practical deployment.

(4) Practical value. I also have an extra major concern on its practical application.In real-world applications, we do not know whether or not the prompts given by the users belong to rare concepts. This method may lead to extra possible degraded quality with inevitable latency disadvantage for general prompts. Note that the experiments of Table 2 on T2I-Bench is not comprehensive enough. Evaluation metrics, such as HPSv2 and PickScore, are more common metrics for general t2i generation.

**Reviewer Scores:**

Reviewer vSy7: 4 -> 6
- Justification: This reviewer raises his score based on the additional experiments and the “theoretical” framing. However, from an AC perspective, their initial concern about the method being “not substantially novel” holds more weight than the rebuttal’s specific fixes.

Reviewer Cm9P: 6 -> 6
- Justification: This reviewer was satisfied with the practical implementation details.

Reviewer jdmt: 6 -> 6
- Justification: This reviewer appreciated the adaptive controller but noted the contribution feels “scoped to a niche use case”. This limitation is critical for the final decision.

Reviewer eyNK: 4 -> 4
- Justification: This reviewer maintained a critical stance, accurately pinpointing that despite the adaptive features, the novelty is limited because “this paper still relies on prompt switching”.

---

### Decision · Program_Chairs · 2026-01-26

Reject